# Pen-grip kinetics in children with and without handwriting difficulties

Yu-Chen Lin[1,2,3], Chieh-Hsiang Hsu[4], Cheng-Feng Lin[5], Hsiu-Yun Hsu[3,6], Jin-Wei Liu[7], Chien-Hsien Yeh[8], Li-Chieh Kuo[2,3,4,8]*

1 Department of Occupational Therapy, Da-Yeh University, Changhua, Taiwan, 2 Institute of Allied Health Sciences, College of Medicine, National Cheng Kung University, Tainan, Taiwan, 3 Department of Occupational Therapy, College of Medicine, National Cheng Kung University, Tainan, Taiwan, 4 Department of Biomedical Engineering, College of Engineering, National Cheng Kung University, Tainan, Taiwan, 5 Department of Physical Therapy, College of Medicine, National Cheng Kung University, Tainan, Taiwan, 6 Department of Physical Medicine and Rehabilitation, National Cheng Kung University Hospital, College of Medicine, National Cheng Kung University, Tainan, Taiwan, 7 Tainan Rehab Clinic, Tainan, Taiwan, 8 Medical Device Innovation Center, National Cheng Kung University, Tainan, Taiwan

☺ These authors contributed equally to this work.
* jkkuo@mail.ncku.edu.tw

**Data Availability Statement:** All relevant data are within the paper and its Supporting Information files.

**Funding:** This work was financially supported by the Medical Device Innovation Center, National

## Abstract

### Introduction

Handwriting difficulty (HD) is a widely discussed issue. Previous researchers have revealed many valuable kinematics related to the handwriting performance. However, a clear understanding of the kinetics of handwriting performance in children with HD is still lacking. Therefore, this study investigated the writing performance of children with HD via a force acquisition pen (FAP), which detects the force applied from the digits and pen tip.

### Methods

Data from 64 school-age children were divided into control (36 children without HD; mean age: 7.97 years) and HD (28 children with HD; mean age: 8.67 years) groups. The participants were asked to perform a tracing task using the FAP at their usual writing pace.

### Results

Compared with the control group, the HD group had significantly less pen-tip force, an average amount of force (in-air) from all three digits, higher force variations (whole task) in the index finger, less force fluctuations with the index and middle fingers and a smaller force ratio.

### Conclusion

The findings of this study suggest that an understanding of the handwriting kinetics and the role of digits in handwriting may be crucial for further planning strategies for handwriting training for children with HD.

 

Cheng Kung University from the Featured Areas Research Center Program within the framework of the Higher Education Sprout Project by the Ministry of Education (MOE) in Taiwan. The authors are also honoured to acknowledge the Ministry of Science and Technology TAIWAN for partially funding this work (MOST 104-2314-B-006 -018 -MY3). The funders had no role in study design, data collection and analysis, decision to publish, or preparation of the manuscript.

**Competing interests:** The authors have declared that no competing interests exist.

## Introduction

Handwriting ability is important to children as they adapt to their school life. Children with handwriting difficulty (HD) may be unwilling to write, unable to finish their homework on time, or have poor self-esteem [1], and some of them may feel physical discomfort, such as muscle fatigue and soreness, during writing activities [2]. The prevalence of HD is between 10% and 34% in school-age children [3, 4]. Typically, handwriting problems are the reason why children with special needs are referred by schools to occupational therapy practitioners [5].

Therapy practitioners use standard assessments to evaluate the performance components related to children's handwriting. These components include kinaesthesia, motor planning, eye-hand coordination, visuomotor integration and in-hand manipulation [6]. In addition to assessing these related skills, some assessments focus on handwriting to describe the writing performance and production based on unique scoring systems, examples of which include the Evaluation Tool of Children's Handwriting [7] and the Minnesota Handwriting Test [8].

After the general evaluation of children's handwriting, the direct intervention of children's grip may be a possible option if children have been determined to have evident gripping problems from a biomechanical perspective. Children may be asked to adapt their grip form and manipulation if necessary. To help therapy practitioners determine how children manipulate a writing utensil, researchers have developed methods to identify different types of grips through observation with the naked eye [9]. With advances in technology, computer-based assessments focusing on handwriting kinematics and kinetics are being commonly used by researchers to understand pen manipulation motion objectively [10–12].

However, the exact amount of force applied from fingers to the writing utensil is difficult to record using the available assessments. For example, therapy practitioners may find that children exert excessive force by observing piercings on the paper, colour change in the fingertip and oral reports from children themselves rather than objectively determining the exact real-time force applied by children. This situation raises a question as to how therapy practitioners should instruct children to apply an appropriate amount of force if the correct amount to be applied remains unknown. Accordingly, a few research groups have attempted to measure the invisible force from fingers with custom-made kinetic pens to present the actual forces applied to the pen [13–16].

Handwriting-related kinetics can be described with three types of forces based on the forces applied from the 'digits' (on the barrel), 'pen' (on the surface)' and 'hand' (on the surface) [17]. Compared with the forces applied from the hand toward a surface, researchers have focused on the forces applied from the digits and pen tip, where therapy practitioners can directly intervene. Previous literature on this topic has suggested that pen-tip force is not grade related [13, 18]. However, it may be influenced by task demands and specific writing characteristics such as writing speed [18]. Some studies have suggested that children with poor handwriting performance exert a relatively low pen-tip force when engaged in specific writing tasks. Rosenblum and Livneh-Zirinski [10] reported that children with developmental coordination disorders (DCDs) apply less pen-tip force compared with typically developing children while performing complex writing tasks. Similar results were found in a study by Chang and Yu [11]. They collected writing data from children with HD, children without HD and children with both DCDs and HD and observed that all of the children applied less pen-tip force when writing complex strokes. Chang and Yu [19] also discovered that children with dysgraphic characteristics applied less pen-tip force in all writing tests at four different complexity levels compared with proficient writers. However, Chau and colleagues [15] showed different results in their work regarding pen-grip activity. They found no difference in the pen-tip force between children with cerebral palsy (CP) and those with typical development.

 

In regard to the grip force applied in children's handwriting, two previous studies suggested that a great grip force is associated with a great tip force, but no specific trend was found in the grip force of a sample of elementary school students at different grade levels [13, 20]. In addition to directly using force to describe the amount of force applied to the act of writing, researchers have suggested the use of different parameters, such as variations in the amount of force, which represents the stability of the force applied over a period of time, and grip-to-tip ratio, which refers to the pattern of force applied from the grip and pen tip, to show how forces behave when applied by digits.

In 2010, Falk et al. indicated that children with non-proficient writing skills show fewer force variations in terms of grip force. Lin et al. [13] enrolled 181 children aged 5–12 years old and observed that the younger ones exhibited more force variations and lower frequency of adjustment in the amount of force applied from each digit. The inconsistent results in the research on force variation are thought to be related to different instructions related to whether a time constraint exists for performing the assigned writing tasks.

Regarding the grip-to-tip ratio, which is the total force applied from the grip divided by the force applied from the pen tip, Lin et al. [13] observed that older children apply force on the pen barrel rather than push the pen tip downward. Similarly, Chau et al. [15] noticed that compared with children with CP, children with typical development generate higher grip-to-tip ratios.

Previous literature has suggested role differences among the thumb, index and middle finger during handwriting [20, 21]. Ghali et al. [22] suggested that each person has a specific and recognisable force distribution on the pen barrel while writing. Shim et al. [23] also proposed a system measuring contact forces from three digits. However, a limited number of studies have reported children's handwriting performance by investigating the forces directly applied from the digits [13, 15, 18, 20, 24–26]. As studies on digit force are still limited, this study was designed as an observational study to provide insights into the issue. This study aimed to ascertain the pen-grip kinetics during writing via the use of a custom force acquisition pen (FAP) system in children with HD compared with those in children without HD. To supplement the motor performance information that may influence handwriting assessments [6, 13], we also recorded fine motor skill performances. Two research questions were asked: (1) Are there any differences in the fine motor skills and pen-grip kinetics between children with and without HD and (2) are there any role differences among the thumb, index and middle finger?

## Methods

### Study design

A cross-sectional design was used to observe handwriting from temporal and kinetic perspectives between children with and without HD. All children received evaluations using the Bruininks–Oseretsky Test of Motor Proficiency, second edition (BOT-2) [27] subtests to survey their fine motor skills. The FAP system was used to acquire the writing kinetics. The Chinese Handwriting Evaluation Form (CHEF) [28, 29] was used to provide additional information related to handwriting problems in the HD group.

### Participants

A total of 36 children without HD (18 girls and 18 boys, mean age 7.97 ± 0.57 years and all right-handed) were recruited as the control group from regular classes in public elementary schools in southern Taiwan. Then, 28 children with HD (13 girls and 15 boys, mean age 8.67 ± 1.42 years; 3 left-handed and 25 right-handed) were recruited from public elementary schools and clinics in urban areas of southern Taiwan. They were referred by parents, teachers,

or therapy practitioners. The mean age of children with HD was more than 8 months older than that of the children without HD. This condition indicated that the children with HD had almost an additional year of handwriting practice compared with their non-HD peers in this study. HDs were identified by oral reports mainly from the participants' teachers and therapy practitioners. Furthermore, the CHEF was given to the HD group. Three subtypes of HD were reported within the HD group: a cognitive learning dysfunction subtype (25%), a motor impairment subtype (28.6%) and a severe hybrid subtype (46.4%). Owing to the small sample size and uneven percentages among the subtypes, the differences between HD subtypes were not analysed. The children and guardians who agreed to participate in this study were asked to sign a consent form approved by the Institutional Review Board at a university hospital. If the children had a history of neurological deficits or severe muscular or orthopaedic problems of the upper extremities, they were excluded from this study.

Although all the children had already been taught to use the dynamic tripod grasp to write in school, the most preferred and frequent manner in which they gripped the pencil was still recorded. The children's pen-grip pattern was recorded as they wrote with a typical pencil. The handwriting grip type classification followed the work of Chang [30]. The sample was divided into five types as follows: a dynamic tripod grasp (HD group: 17.9%, control group: 25.7%), a lateral tripod grasp group (HD group: 50%, control group: 48.6%), a quadruped grasp group (HD group: 7.1%, control group: 5.7%), a lateral quadruped grasp (HD group: 25%, control group: 14.3%) and others (HD group: 0%, control group: 5.7%).

## Instruments

**FAP system.** The handwriting kinetics were collected using the FAP system (Fig 1). The FAP was designed as a common ball-point pen that contains three thin-beam force sensors and one button-shaped transducer (TBS-5 and SLB-50, Transducer Techniques, Temecula LLC, CA, USA) to simultaneously detect the forces applied from three digits (thumb, index and middle fingers) and the pen tip, respectively. The positions of the three force sensors on the FAP were adjustable. The FAP weighed about 30 g and was similar in appearance to a typical pen (length of pen: 12 cm; and diameter of shank: 1.1 cm). The force data were acquired and converted with an instruNet network device (iNet-100, GW Instruments, Inc., MA, USA). The sampling rate for the force data acquisition was set at 70 Hz. A low-pass filter was applied using MATLAB programmes to remove noise, with a cut-off frequency of 6 Hz (MathWorks Ltd., Natick, MA, USA).

The participants were asked to trace numbers (0–9) on a piece of A4-size paper using the same FAP at their natural writing speed. To avoid interference due to variations in character size, grey marks were printed on the writing sheet for tracing. In addition, the sequence of each stroke was fixed with visual guidance on the writing sheet.

**BOT-2.** The fine motor skills were evaluated using the BOT-2 subtests related to fine motor skills, including 'fine motor precision', 'fine motor integration' and 'manual dexterity' [31]. The BOT-2 is a standardised, norm-referenced measurement with good test-retest reliability (ICC = 0.99, 95% confidence interval) and internal consistency (Cronbach's $\alpha$ = 0.92) [27]. To compare the participants' performance fairly, we used the raw scores for the three subtests to conduct a statistical analysis rather than the standard scores, which are controlled by age.

**CHEF.** The information related to handwriting problems was identified using the CHEF [28]. CHEF is a questionnaire with 25 items comprising five major dimensions, including construction, accuracy, speed, grip ergonomics and directionality. Children with handwriting deficits can be classified into five subtypes: mild, moderate and severe hybrid, motor impairment

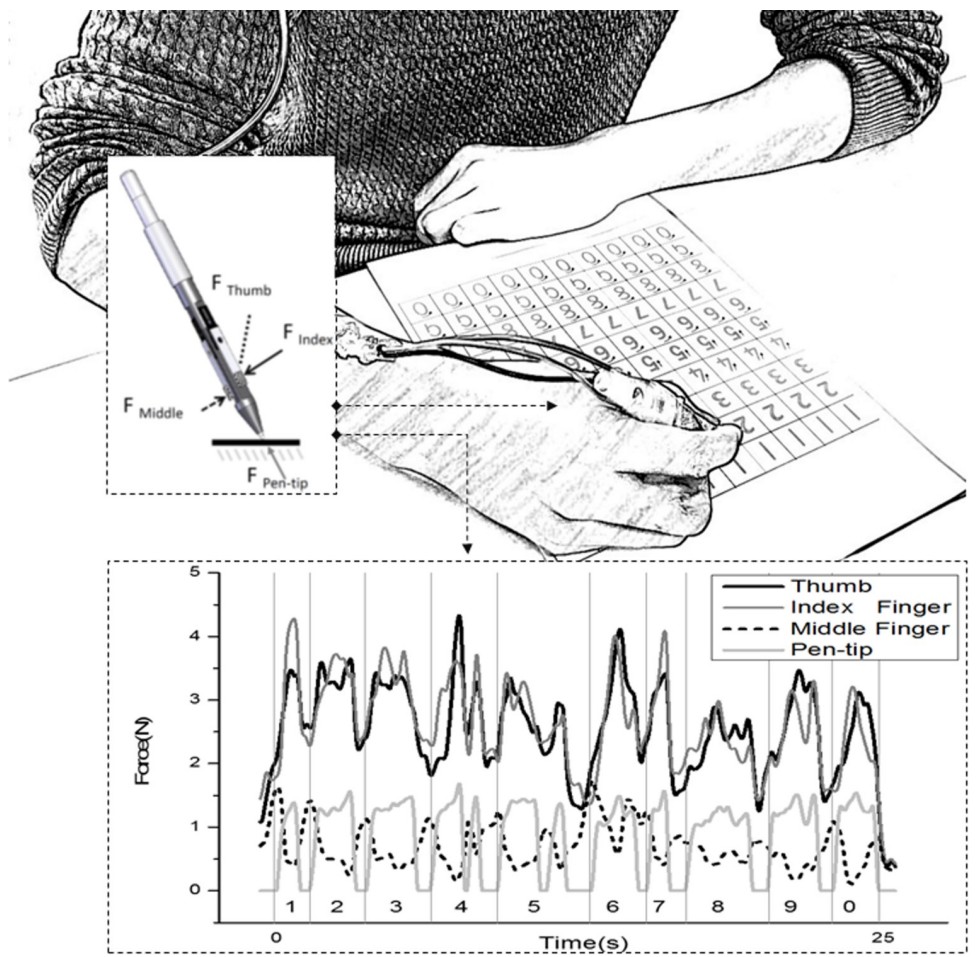

**Fig 1. FAP system.** Settings of the FAP system, pen design and force data obtained in one trial from the control group.

and cognitive learning dysfunction [29]. In accordance with the manual, children classified into the mild hybrid subtype mainly have deficits in the grip ergonomics dimension. Opposite the mild hybrid subtype, children classified into the moderate hybrid subtype show unobvious deficits in the grip ergonomics dimension and a small amount of deficits in the motor impairment or cognitive learning dysfunction dimension. The severe hybrid subtype is the most serious subtype, reflecting severe deficits in all five dimensions and poor perceptual motor ability. The motor impairment subtype is mainly characterised with deficits in writing speed, construction of characters, DCDs, writing automation and fine motor control. Children classified into the cognitive learning dysfunction subtype lack appropriate, efficient learning strategies.

## Procedures

The CHEF was provided to the participants' parents, teachers, or therapy practitioners for the HD group. All participants in both groups were asked to complete the tracing tasks with the FAP system and take the BOT-2 subtests in a random order. Although the positions of the three force sensors on the FAP were adjustable, no participant in this study, however, requested the examiner to adjust the position of the sensors due to discomfort or an awkward hand posture when the examiners asked them if they needed to adjust the position of sensors.

They were requested to trace numbers in each trial, with three trials each. They had a 1 min rest interval between each trial and sufficient time to practice using the FAP with a tripod grasp before the formal trials. Each participant sat in front of a height-adjustable desk with both arms placed comfortably on it, with a 90˚ knee flexion and feet resting on the floor. The entire process was completed in 1 h for each participant.

## Handwriting performance parameters

Four kinetic and one temporal parameters, including average force (AF) [32, 33], task time (TT), coefficient of variations in force (CVF) [14], the number of force fluctuations per second (NFFPS) and the force ratio (FR) [15], were used to describe the children's handwriting performance. Table 1 describes the details and clinical meanings of all of the parameters.

Handwriting performance consists of two components, namely, in-air and on-paper components. The in-air component has been reported in some studies related to children's handwriting [10, 16, 34]. The finding of these studies suggested that children with non-proficient handwriting or DCDs have longer in-air time compared with proficient handwriters or typically developing children. Drotár et al. [35] suggested that in-air movement during handwriting may involve cognitive processes, such as motor planning and programming of motor sequences. Therefore, according to the pen-lifting and pen-touching status, the parameters were set to different conditions, where 'Whole task' represented the data calculated from the initial pen-touching action to the last pen-lifting action in one trial. 'On paper' implied that the data were calculated while the pen tip was touching the paper in one trial. 'In air' indicated that the data were calculated while the pen tip was not touching the paper in one trial.

The parameters were computed using custom MATLAB programmes. All parameter data were averaged (over three trials) for further statistical analyses.

**Table 1. Definition, computation and clinical meanings of the five kinetic parameters.**

| Parameters | Definition | Computation details | Clinical meanings |
|---|---|---|---|
| Average force (AF) | The mean force for one trial. | Four AFs were obtained, with the AF from the pen tip being computed under only one condition, which is 'On paper'. | A high AF may lead to pale knuckles, pain or piercing of the paper. Conversely, a low AF might lead to deposition of a small amount of ink on the paper and poor writing quality. |
| Task time (TT) | The total time from the first pen-touching action to the last pen-lifting action in one trial. | The TT was set using the force data measured from the pen tip. The initiation time was defined as the first time at which the force value was 0.1 N higher than the baseline, which was the AF during the initial 0.5 s. The termination time was defined as the last time point at which the force value was 0.1 N higher than the baseline. | TT indicates the writing speed. |
| Coefficient of variation in force (CVF) | The ratio of the standard deviation (SD) of force over the AF in one trial. | The measures were obtained from the computation of the SD of force over the AF. | A high CVF indicates a dynamic force output from the digits or the pen tip. |
| Number of force fluctuations per second (NFFPS) | The total number of positive peaks of force divided by the total TT in one trial. | In one trial, the total number of positive peaks of force obtained from the thumb can be more than 20, based on the number of strokes and the frequency with which they changed direction. When the NFFPS increases, the frequency of pushing and pulling from the three digits also increases. | The greater the NFFPSs, the more frequent were the adjustments in force exertion during writing. |
| Force ratio (FR) | The force of the pen tip divided by the total force of the three digits. | The FR was computed when the pen was touching the paper. | The grip-to-normal force ratio represents the pattern of energy expended by the hand engaged in the pen-grip activity [15]. |

## Statistical analysis

The statistical analysis was performed using SPSS 17.0 (Statistical Package for Social Sciences Inc. Chicago, IL, USA). Descriptive statistics were used to calculate the means and standard deviations (SDs) of the demographic data. An analysis of covariance (ANCOVA) controlling for age was used to analyse the experimental data to determine the between-group differences in the parameters ($AF_{on\ paper}$ of the pen tip, TTs, $CVF_{on\ paper}$ of the pen tip and FR) and raw BOT-2 scores. A mixed-model ANCOVA, including one between-group factor (group: with HD vs. without HD) and one within-group factor (role of the digits: thumb vs. index finger vs. middle finger), was performed on the $AF_{on\ paper}$, $AF_{in\ air}$, $AF_{whole\ task}$, $CVF_{whole\ task}$ and $NFFPS_{whole\ task}$, with age as a covariate. Bonferroni tests were used as post-hoc tests. The level of significance was set at $p < 0.05$.

## Results

Table 2 shows the means and SDs for all parameters and fine motor scores.

Table 2 also shows the ANCOVA results on the group differences for the parameters ($AF_{on\ paper}$ of the pen tip, TTs, $CVF_{on\ paper}$ of the pen tip and FR) and fine motor scores. For the fine motor skills, scores for 'fine motor precision' and 'fine motor integration' were significantly lower in the HD group. The $AF_{on\ paper}$ of the pen tip was significantly lower in the HD group. For the TTs, between-group significance was only found in the $TT_{whole\ task}$. For the $CVF_{on\ paper}$ of the pen tip, no significance was found between groups. Finally, the $FR_{on\ paper}$ was significantly lower in the HD group.

Table 3 shows the results for the effects of group and digit type on the $AF_{in\ air}$, $AF_{on\ paper}$, $AF_{whole\ task}$, $CVF_{whole\ task}$ and $NFFPS_{whole\ task}$ via the mixed-model ANCOVA controlling for age. No interaction effect was observed between the group and digit type when analysing the $AF_{in\ air}$, $AF_{on\ paper}$, $AF_{whole\ task}$ and $NFFPS_{whole\ task}$. An interaction effect was found between the group and digit type when analysing the $CVF_{whole\ task}$. Therefore, the simple main effect was analysed with ANCOVA (for the group) and one-way analysis of variance (ANOVA) (for the digit type) for $CVF_{whole\ task}$. For the AFs and $NFFPS_{whole\ task}$, no significant difference among digit types was found in the $AF_{in\ air}$, $AF_{on\ paper}$ and $AF_{whole\ task}$ or $NFFPS_{whole\ task}$. The scores for $AF_{in\ air}$, $AF_{on\ paper}$, $AF_{whole\ task}$ and $NFFPS_{whole\ task}$ were significantly greater in the control group. For the $CVF_{whole\ task}$, the ANCOVA (for the group) result showed that only the $CVF_{whole\ task}$ of the index in the HD group was significantly greater than that in the control group (difference between groups in three digit types [thumb], F = 1.13, $p = .29$, $\eta_p^2 = .018$; [index], F = 20.30, $p = .00$, $\eta_p^2 = .250$; [middle], F = 0.41, $p = .53$, $\eta_p^2 = .007$). The one-way ANOVA (for the digit type) result showed significant differences in HD (difference among digits, $F = 6.61$, $p = .00$, $\eta_p^2 = .197$) and control (difference among digits, $F = 5.29$, $p = .02$, $\eta_p^2 = .131$) groups. In the control group, the post-hoc tests showed that the $CVF_{whole\ task}$ of the thumb and middle finger were significantly higher than those of the index finger (thumb vs. index finger, $p = .02$; thumb vs. middle finger, $p = .43$; index finger vs. middle finger, $p = .03$). In the HD group, the $CVF_{whole\ task}$ of the index finger was significantly higher than that of the thumb (thumb vs. Index finger, $p = .01$; thumb vs. middle finger, $p = .43$; index finger vs. middle finger, $p = .14$).

## Discussion

HD is a wildly discussed issue that has received considerable attention from clinical and educational professionals. Previous studies have revealed valuable kinematics related to handwriting performance. However, the reports on children's handwriting performance by investigating the force directly applied from the digit are still limited. This study discovered the pen-grip

**Table 2. Descriptive statistics data in two groups and the ANCOVA results for between-group difference.**

| Kinetics of handwriting (sources of force) | Group | | | | | |
|---|---|---|---|---|---|---|
| | Control (n = 36) | | HD (n = 28) | | | |
| | Mean | SD | Mean | SD | Sig.$^a$ | $\eta_p^{2\,a}$ |
| Fine motor skills | | | | | | |
| Fine motor precision | 38.44 | 2.16 | 33.07 | 4.95 | .000** | .473 |
| Fine motor integration | 37.22 | 2.04 | 33.86 | 5.67 | .000** | .222 |
| Manual dexterity | 25.94 | 2.91 | 25.79 | 4.98 | .128 | .037 |
| Average Force (AF), unit: Newton | | | | | | |
| Thumb -on paper | 3.07 | 1.31 | 2.68 | 0.64 | | |
| Thumb -in air | 1.56 | 0.87 | 1.12 | 0.38 | | |
| Thumb -whole task | 2.70 | 1.22 | 2.27 | 0.53 | | |
| Index -on paper | 2.11 | 0.77 | 1.75 | 0.63 | | |
| Index -in air | 1.15 | 0.55 | 0.67 | 0.26 | | |
| Index -whole task | 1.87 | 0.72 | 1.51 | 0.53 | | |
| Middle -on paper | 1.34 | 0.75 | 1.09 | 0.56 | | |
| Middle -in air | 1.05 | 0.49 | 0.75 | 0.32 | | |
| Middle -whole task | 1.29 | 0.69 | 1.07 | 0.53 | | |
| Pen Tip -on paper | 0.95 | 0.29 | 0.77 | 0.36 | .023* | .082 |
| Task Time (TT) | | | | | | |
| TT -on paper (ratio) | 0.71 | 0.08 | 0.72 | 0.05 | .635 | .004 |
| TT -in air (ratio) | 0.29 | 0.08 | 0.28 | 0.05 | .664 | .003 |
| TT -whole task (s) | 23.46 | 7.49 | 27.12 | 10.05 | .040* | .067 |
| Coefficient of Variation in Force (CVF) | | | | | | |
| Thumb -whole task | 0.38 | 0.09 | 0.40 | 0.08 | | |
| Index -whole task | 0.35 | 0.08 | 0.48 | 0.15 | | |
| Middle -whole task | 0.42 | 0.14 | 0.43 | 0.12 | | |
| Pen Tip -on paper | 0.36 | 0.05 | 0.34 | 0.06 | .321 | .016 |
| Number of force Fluctuations per second (NFFPS) | | | | | | |
| Thumb -whole task | 1.34 | 0.24 | 1.26 | 0.17 | | |
| Index -whole task | 1.36 | 0.17 | 1.26 | 0.22 | | |
| Middle -whole task | 1.59 | 0.38 | 1.40 | 0.25 | | |
| Force Ratio (FR) -on paper | 0.16 | 0.05 | 0.11 | 0.06 | .002** | .147 |

*Note.* SD = standard deviation; HD = handwriting difficulty

*$p < 0.05$

**$p < 0.01$.

$^a$ Results of the ANCOVA correcting for age as a covariate

kinetics during writing via the FAP system of children with HD compared with those without HD. In addition, the relationship among three digits during writing was discussed to better understand the digits' role.

## Group differences

Based on the demographic and measured data for the two groups, although the mean age of the HD group was higher than that of the control group, which might have indicated the better motor performance of this older group, the fine motor precision and integration of the HD group were poorer than those of the control group. In addition, no between-group differences were found in manual dexterity. This result was consistent with the argument that poor

**Table 3. ANCOVA summary table for the effects of the digit, group and their interaction.**

| Kinetics of handwriting | Covariance (Age) | | | Group | | | Digit | | | Group ⊆ Digit | | | |
|---|---|---|---|---|---|---|---|---|---|---|---|---|---|
| | *F* | *Sig.* | $\eta_p^2$ | *F* | *Sig.* | $\eta_p^2$ | *F* | *Sig.* | $\eta_p^2$ | *F* | *Sig.* | $\eta_p^2$ | Post-hoc |
| $AF_{on\ paper}$ | 0.82 | .67 | .01 | 4.85 | .03* | .07 | 1.97 | .14 | .03 | 0.21 | .81 | .00 | Ct > HD |
| $AF_{in\ air}$ | 1.15 | .29 | .02 | 13.38 | .00* | .18 | 0.76 | .47 | .01 | 0.69 | .50 | .01 | Ct > HD |
| $AF_{whole\ task}$ | 1.06 | .31 | .02 | 5.95 | .02* | .09 | 1.49 | .23 | .02 | 0.56 | .57 | .01 | Ct > HD |
| $CVF_{whole\ task}$ | 2.40 | .13 | .04 | 7.82 | .01* | .11 | 0.58 | .54 | .01 | 7.33 | .00** | .11 | Interaction[a] |
| $NFFPS_{whole\ task}$ | 0.39 | .53 | .01 | 5.89 | .02* | .09 | 1.75 | .19 | .03 | 1.58 | .21 | .03 | Ct > HD |

*Note.* AF, average force; CVF, coefficient of variation in force; NFFPS, number of force fluctuations per second; Ct, control group; HD, handwriting difficulty group.

Post-hoc type: Bonferroni

*$p < 0.05$

**$p < 0.01$

[a] Simple main effect(showing the significant results):

ANCOVA for group, $CVF_{index}$: HD > control ($p = .00$)

One-way ANOVA for digit type, CVF, HD: index finger > thumb ($p = .01$), control: thumb > index finger ($p = .02$); middle finger > Index finger ($p = .03$)

handwriting is related to poor eye-hand coordination, visual motor integration, or in-hand manipulation [6, 36]. However, the data in this study were analysed in children with uneven percentages of distribution among all three subtypes of HD. As HD is very complex, and the assessment of HD classification has only been developed recently, the results obtained in this study should be interpreted with caution when generalise the findings to children with different types of HD.

The results showed that the AF from the pen tip was lower in the HD group, which is consistent with previous reports showing that children with dysgraphic characteristics [11, 19] or poor motor ability [10, 15] may apply less pen-tip force. These results may be explained by considering the difficulty that children with HD have in regard to maintaining stability when increasing the pen-tip force to exploit the degree of friction between the pen and the writing surface [12]. Nevertheless, the average between-group difference in AF from the pen tip was 0.18 N, which is rarely observed in real situations. Further, the AFs from different digits were evaluated in three different pen positions (whole task, on-paper and in-air). Moreover, in all positions, the AF from all digits was significantly lower in the HD group. Based on the result, loosening the fingers when lifting the pen may influence children's writing performance. When the pen is picked up, the constantly applied force from the digits is unnecessary due because the ink is not made on paper. Constant force application may be unnecessary from the energy-saving perspective. On the other hand, it may reduce the frequency to enlarge the applied force. One possibility is that these children could not generate the force constantly. Another possibility is that these children do not know how to constantly apply force. However, more evidence is needed to confirm this hypothesis, such as the muscle endurance test or skill check test. To the best of our knowledge, no studies reported the applied digit force during pen lifting. This study may thus be the first to report these data based on the handwriting performance of children. However, researchers have discussed the kinematic and temporal variables that occur during writing movements.

Chang and Yu [19] reported that children with dysgraphic characteristics exhibited a higher in-air to on-paper time ratio compared with a control group in some writing tasks. Rosenblum et al. [16] observed that children with poor handwriting had a lower in-air to total time ratio and a lower speed. Our results showed no significant between-group differences for $TT_{in\ air}$ and $TT_{on\ paper}$. However, the results for the $TT_{whole\ task}$ showed that the children with HD

needed more time compared with the control group, which concurred with the finding of Rosenblum et al. [16]. In a study conducted by Kushki et al. [32], children with dysgraphia wrote faster than those who are proficient at writing, which may be due to a speed-accuracy trade-off. The reason for the inconsistency of time may have been the different time constraints and writing tasks requirements among the examined studies. In our study, the writing time was not constrained. When children in both groups needed to complete a tracing task with similar accuracy, our results showed that children in the HD group needed more time to complete the task. Similarly, the difference in the findings of $TT_{in\ air}$ and $TT_{on\ paper}$ may also have been due to the effect of different time constraints and writing task requirements.

A low CVF represents a static way of grasping a pen in terms of the amount of force. Although not all CVFs showed significant between-group differences, the mean CVF values for the three digits were smaller in the HD group. Our results indicated that the children with HD used a less stable method by which to control the amount of force applied to manipulate the pen during writing. This result was consistent with that of a previous study, where the researchers observed that children with mature motor abilities used a static pattern to manipulate the pen in terms of the amount of force applied [13].

In this study, NFFPS, a novel parameter, was also used to describe the frequency of force adjustments. The results showed that the NFFPSs of all three digits were higher in the control group. The results were consistent with those of a previous study [13], where children with mature motor abilities exhibited higher adjustment frequencies. This result indicates that although proficient writers grip the pen in a static way in terms of the amount of force, they may still use very rapid force adjustments to manipulate the pen.

The concept of open-loop and close-loop motor control have been applied in several studies focusing on handwriting movement [11, 35, 37]. A more automated movement is executed as an open-loop and feed-forward control movement. One explanation for the different writing patterns between children with and without HD is that feedback-controlled movements have been found in children with HD. This condition suggests that the unstable amount of force exerted resulted from the processing of feedback-controlled movements, followed by less frequent adjustments. In addition, the relatively low force applied while pen lifting in the HD group can possibly be related to this feedback-control process. However, more future research is needed to support this viewpoint.

The literature shows that children with better motor abilities apply more force to the pen barrel than to the pen tip [13, 15]. However, the present study showed the opposite results for the FR. The FR in the HD group was significantly lower, which means that the HD group used more effort to manipulate the pen rather than press it downward. The most likely explanation for this result rests in the specific characteristics of the grip pattern of these children. Compared with a previous study [13], where writing data from 170 typically developing children were surveyed and where the mean FR value from kindergarten to sixth grade was between 0.17 and 0.14, the FR (0.11±0.06) in the HD group in the present study was not in this range. Nevertheless, as these studies [13, 15] evaluated different target groups (HD/ CP/ children without HD), future research is evidently required to supplement the data with related target subjects.

Given that handwriting is complex, the reasons for the resulting HD vary. Therefore, this study focused on one concept, that is, the writing kinetics, to reveal the possible and measurable factors related to children with HD. For example, asking children to strengthen or loosen the force applied from a specific digit is a relatively direct way to change the writing pattern. However, in the real world, we could not know if this step is necessary or whether asking them to change the force is correct if we do not know the amount of force considered as poor. Therefore, this study provided real force data, such as the AF value, suggested more critical

situations for observing, such as pen in air, and proposed the particular digit for observing such as index finger to tentatively detect the possible and manipulatable factors related to HD. In addition, this study introduced CVF, NFFPS and FR to explain the force control pattern in children with HD. The study provided a practical method for the teachers and clinicians to assess children's handwriting problems based on writing kinetics.

## Role differences

To determine the role of each digit on pen grip, we used three parameters (AF/CVF/NFFPS) in the control and HD groups. Although, the results of the AFs showed no significant difference among digits, the mean value of the AFs still showed a similar trend (thumb > index finger > middle finger) in both groups (Table 2), which was consistent with the previous literature [13, 21]. However, the results of the CVFs were inconsistent in the two groups. The CVF of the index finger was considerably higher among the three digits in the HD group. This result may have been due to the extremely low value of the $AF_{in\ air}$ of the index finger in the HD group. One possible explanation for this was that children with HD may be unable to maintain the same pen-grip pattern during writing. This condition may result in their extending the index finger to release the pen. Meanwhile, the thumb and middle finger do not loosen the pen grip to avoid dropping the pen. Although previous studies have shown that some children change their grip pattern during writing, no evidence proved that more children with HD change their grip patterns while writing compared with children without HD [2, 38, 39]. On the contrary, the CVF of the index finger was lower than that of the thumb and middle finger in the control group. However, unlike the HD group, neither extreme data nor specific observations were found in the control group. Although the reason for the low CVF value for the index finger in the control group remains unclear, we could not ignore that the motion of the index finger in the HD group can be described based on the AF and CVF. Future research may be necessary to explore this phenomenon by acquiring related kinetic parameters or additional kinematic information.

No significant differences were observed among the three digits based on the NFFPSs. However, the mean value of the NFFPSs still exhibited the same trend in both groups (middle finger > thumb, index finger). This trend was generally compatible with results of a previous study [13] that tested the number of force fluctuations and suggested that the middle finger may be crucial to the degree of control carried out by the other two digits.

In conclusion, the highest amount of AF from the thumb suggested that it may play a steering role. In addition, the highest NFFPS value from the middle finger suggested that the ability of the middle finger may be crucial to the control carried out by the other digits.

## Limitations and future studies

The first limitation in this study concerns the identification of HD used. The CHEF is widely used to assess handwriting problems in Taiwan. However, as it is a problem-based questionnaire and lacks the capability to directly score a handwriting product based on standard samples, additional measurements to evaluate handwriting should be considered. Second, the findings were limited to the tracing task used in this study and may not be directly applicable to actual writing situations. A future study can measure actual writing tasks, such as free writing or copying. Third, the tasks assigned in this work were designed to record a very natural pattern generated by children. Thus, between-group differences might have been easily induced. A future study may add a time constraint to address handwriting kinetics under different circumstances. Fourth, the FAP system can only measure a tripod grasp. Although all participants in this study can use a tripod grasp to grip the FAP, about 30% of them used a

non-tripod grasp with a typical pen. Thus, the original grasp pattern may have affected the kinetic data. Further, a FAP that can detect grip kinetics from different pen grip patterns other than the tripod grasp is still needed. Fifth, the current study only examined the kinetics of handwriting. A kinematic method should be designed to analyse the motion used for handwriting in general. Finally, although children with three subtypes of HD were reported according to the CHEF classification, this study did not analyse the differences in the characteristics of writing kinetics among the various HD subtypes due to the small sample size and uneven percentages of distribution among the subtypes in the current study sample. A future study with a larger sampling scale is suggested to compare the differences in the writing features among HD subtypes.

## Conclusion

In this study, handwriting kinetics were analysed using parameters describing the control of digits and exploring the differences between children with and without HD. On the basis of the results, our findings suggested that when intervening in children's handwriting problems, children loosen the gripping digits despite the digits appearing to touch the pen barrel during writing, especially at the moment when they lift the pen to write the next stroke.

Digit role differences can also be considered when evaluating writing. Compared with the children with HD, the children without HD used a more static method by which to apply the force required to grip the pen and generated faster force adjustments to manipulate the pen from a kinetic point of view. Despite its limitations, this study still provides a number of insights into force control for clinicians and teachers in formulating evaluations and interventions for HD.

## Supporting information

**S1 Table. Summary table of the parameters for the three subtypes of HD group.**
(DOCX)

## Acknowledgments

We thank the students, parents, teachers and clinicians who participated in the data collection for this study. We are grateful to Dr. Chung-Ying Lin for providing the statistical consulting services from the Biostatistics Consulting Center, Clinical Medicine Research Center, National Cheng Kung University Hospital.

## Author Contributions

**Conceptualization:** Yu-Chen Lin, Cheng-Feng Lin, Hsiu-Yun Hsu, Chien-Hsien Yeh, Li-Chieh Kuo.

**Data curation:** Yu-Chen Lin, Chieh-Hsiang Hsu, Jin-Wei Liu.

**Formal analysis:** Yu-Chen Lin, Chieh-Hsiang Hsu, Cheng-Feng Lin, Hsiu-Yun Hsu.

**Investigation:** Yu-Chen Lin, Jin-Wei Liu.

**Methodology:** Yu-Chen Lin, Li-Chieh Kuo.

**Project administration:** Yu-Chen Lin.

**Resources:** Yu-Chen Lin, Chieh-Hsiang Hsu, Hsiu-Yun Hsu, Jin-Wei Liu, Chien-Hsien Yeh, Li-Chieh Kuo.

**Software:** Yu-Chen Lin, Chieh-Hsiang Hsu.

**Supervision:** Yu-Chen Lin, Cheng-Feng Lin, Hsiu-Yun Hsu, Li-Chieh Kuo.

**Validation:** Yu-Chen Lin, Chieh-Hsiang Hsu.

**Visualization:** Yu-Chen Lin, Chieh-Hsiang Hsu.

**Writing – original draft:** Yu-Chen Lin, Li-Chieh Kuo.

**Writing – review & editing:** Yu-Chen Lin, Li-Chieh Kuo.

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
