## [Decision Letter · Decision Letter 0]

2 Jun 2021

PONE-D-21-12263

Pen-grip Kinetics in Children with and without Handwriting Difficulties

PLOS ONE

Dear Dr Li-Chieh Kuo,,

Thank you for submitting your manuscript to PLOS ONE. After careful consideration, we feel that it has merit but does not fully meet PLOS ONE’s publication criteria as it currently stands. Therefore, we invite you to submit a revised version of the manuscript that addresses the points raised during the review process.

We look forward to receiving your revised manuscript.

Kind regards,

Victor Frak, MD, Ph.D

Academic Editor

PLOS ONE

Journal Requirements:

Reviewers' comments:

Reviewer's Responses to Questions

**Comments to the Author**

1. Is the manuscript technically sound, and do the data support the conclusions?

Reviewer #1: No

Reviewer #2: Partly

2. Has the statistical analysis been performed appropriately and rigorously? 

Reviewer #1: Yes

Reviewer #2: No

3. Have the authors made all data underlying the findings in their manuscript fully available?

Reviewer #1: Yes

Reviewer #2: Yes

4. Is the manuscript presented in an intelligible fashion and written in standard English?

Reviewer #1: No

Reviewer #2: Yes

5. Review Comments to the Author

Reviewer #1: The article entitled “Pen-grip Kinetics in Children with and without Handwriting Difficulties” describes an experimental research. Children with and without handwriting difficulties (HD) were evaluated by means a pen with force sensors while writing numbers in a previously printed in grey sheet without any other constraints. The force applied by the three fingers – thumb, index and middle – was recorded along the period when the pen was in touch with the sheet and in the air.

The Introduction Section was well designed, and the lecture easily flows toward the objectives presented at its end. However, only one article cited in this section was published in the last five years and nine from the 23 of them were published in the last century. This selection makes us wonder the relevance of the research object and the addition that the results to be obtained can bring to what is already described in the literature.

This wondering rises when the sample is detailed. Three HD forms are found in the sample: “a cognitive learning dysfunction subtype (25%), a motor impairment subtype (28.6%), and a severe hybrid subtype (46.4%).” Further on, the text describes each of these types of HD: “The severe hybrid subtype is the most serious subtype, reflecting severe deficits in all five dimensions and poor perceptual motor ability” ... “The motor impairment subtype is mainly characterized with deficits in writing speed, construction of characters, DCD, writing automation, and fine motor control” … “Children classified into the cognitive learning dysfunction subtype lack appropriate, efficient learning strategies”. In this sense, the differences between each subtype are significant, either by severity or specificity. However, the sample is treated as a single group so that there can be statistical strength. Finally, at the beginning of the discussion, when commenting on the homogeneity of manual dexterity between the groups, it is stated that “this might be related to the fact that most of the writing difficulty subtypes were “motor impairment” and “severe hybrid,” which are both related to poor motor performance during handwriting”, which demonstrates an important reductionism of the type "severe hybrid". This statement also puts a quarter of the sample out of the discussion by assuming that the argument used for both of that types would not apply to the "cognitive learning dysfunction" type.

In addition, the study observed five different categories of handwriting grip with different intra-group and – mainly – between-groups distributions. There is no mention of the distribution of these categories of handwriting grip in the three types of HD in the sample.

There is an important phrase related to this topic whose wording is ambiguous in my opinion:

“Although the positions of the three force sensors on FAP are adjustable, in this study, all participants used the same position setting for the force sensors based on their own grip.”

What does it mean? That all children used the same position setting of the sensors in the pen, regardless of how each child grasped it? Or that each child had an individualized configuration of sensors according to their form of grasping? Or, yet, that there were several configurations, one for each form of handwriting grasping, but not one for each child according to their natural positioning?

This questioning is important because it can indicate whether there was an extra constraint factor for the HD group. If the positioning was unique, its effects on the HD group may have been greater than on the group without HD. If the positioning was determined by the way of grasping, how many subgroups were formed? And if the positioning was individualized by a child, what did you mean by "same position setting"?

Therefore, I believe that, by not adequately considering the heterogeneity of the HD sample, the authors compromised the reliability of the results. As stated by them, “due to the small sample size and uneven percentages among the subtypes, differences between HD subtypes were not analyzed”.

Some minor considerations:

The model of direct citations used in the article (citing only the first two authors) is not adequate, since it gives a single comma the power to indicate that there are other authors besides the two mentioned. Forms such as "Author and colleagues", "Author and collaborators" and even "Author et al." are preferable in these cases.

Although it could be considered as implied, the text does not formally indicate whether the sample is composed only by right-handed children and does not comment on the consistency of manuality. They belong to a key age range regarding manuality definition, and the topic deserves to be considered in the characterization of the sample (Scharoun & Bryden, 2014).

The Cronbach’s alpha value for the BOT-2 test is 0.92; however, ideal values for Cronbach’s alpha range from 0.7 to 0.9. Values greater than 0.9 indicate test redundancy (Streiner, 2003).

Right following the phrase “There have been few studies reporting on applied force while pen-lifting” there are no references to these few studies, nor a discussion comparing the results obtained in the study with this literature, which is very relevant here.

Scharoun, S. M. and Bryden, P. J. Hand preference, performance abilities, and hand selection in children. Front Psychol. 2014; 5: 82. doi: 10.3389/fpsyg.2014.00082

Streiner, D. L. Being inconsistent about consistency: when coefficient alpha does and doesn’t matter. Journal of Personality Assessment. 2003; 80:217-222. doi: 10.1207/S15327752JPA8003_01

Reviewer #2: The aim of this study was to characterise the pen-grip kinetics profile of children with and without handwriting difficulties (HD), an issue that has received little attention to date. Despite the relevance of the topic, certain aspects limit the impact of the study, for example, the absence of an in-depth analysis of past studies; the use of a heterogeneous sample of children with handwriting difficulties; and the statistical approach to data analysis. The points that need to be clarified and strengthened are detailed below.

Introduction

The introduction requires a more detailed review of previous studies. According to the authors, “the results of previous studies related to children’s handwriting kinetics remained inconsistent” (p.7, lines 2-3), meaning that one of the main motivations of the present study was to provide more evidence in regard to the issue. According to the above, the reader would expect to be provided with possible explanations for the heterogeneity of previous results. However, only in the case of findings concerning variations in terms of grip force were certain methodological reasons mentioned as a source of inconsistency (p.6, lines 13-15). A more in-depth analysis of previous findings would make for a more compelling argument as to the specific contribution of the present study.

The research questions overlap to a certain degree. It would be advisable to reformulate them, as this would also help to improve the statistical analysis strategy.

Method

My main concerns in this section are in regard to the sample. The group of children classified as having HD is highly heterogeneous according to the profiles extracted from the Chinese Handwriting Evaluation Form (CHEF), so it is difficult to understand the implications of the differences found later with regard to the control group. The difference in the age range between the groups should also be emphasised, as the children with HD had almost an additional year of handwriting practice compared to their non-HD peers, a fact which could explain the lack of differences in some measures.

The pen-grip patterns were not distributed in the same way between the groups with and without HD. The lateral quadruped grasp was more prevalent within the HD group, while the dynamic tripod grasp was more prevalent within the control group. Do you think this could be a source of explanation for the differences found between the groups? This point is especially relevant if we consider that, as I understand it, the positions of the three force sensors were fixed with a tripod grasp pattern: “Although the positions of the three force sensors on FAP are adjustable, in this study, all participants used the same position setting for the force sensors based on their own grip. They were requested to trace numbers in each trial, with three trials each. They had a one-minute rest interval between each trial and enough time to practice using the FAP with a tripod grasp before the formal trials” (p.10, lines 18-22).

Results

This section requires a major revision.

The analysis strategy used to answer the two questions could be improved. To answer research question one (Are there any differences in the fine motor skills and pen-grip kinetics between children with HD and without HD?), a series of t-tests were used to compare the groups using variables that are correlated; this is not a good idea given the increased chance of a type I error. The differences in age should have been controlled.

To answer research question two (Are there any differences in pen-grip kinetics among different digits that indicate corresponding roles?), three one-way ANOVAs were used for each group. At this point, one wonders why the authors were not interested in the differences between the groups, although this is related to the problem of overlap in the research questions. As the comparison between groups was not of interest to the authors, they compared the force grip of the three digits for each group and parameter. A less complex and more robust approach could have been achieved using a mixed model.

Discussion

Given that the main argument of the study was based on the inconsistencies in the findings of previous studies, I would have expected a discussion reporting explicitly on how the results of the present work help to clarify these inconsistencies. However, the discussion does not address this.

The authors appropriately highlight the limitations of the work, but do not address the practical implications that the results could have for the identification and intervention of children with HD.

6. PLOS authors have the option to publish the peer review history of their article (what does this mean?). If published, this will include your full peer review and any attached files.

Reviewer #1: **Yes: **Ronaldo Luis da Silva

Reviewer #2: No

---

## [Author Response · Author response to Decision Letter 0]

11 Aug 2021

Reviewer #1: 

The article entitled “Pen-grip Kinetics in Children with and without Handwriting Difficulties” describes an experimental research. Children with and without handwriting difficulties (HD) were evaluated by means a pen with force sensors while writing numbers in a previously printed in grey sheet without any other constraints. The force applied by the three fingers – thumb, index and middle – was recorded along the period when the pen was in touch with the sheet and in the air.

We sincerely appreciate the Reviewer’s thorough review and valuable suggestions. The authors revised the manuscript according to your constructive comments. The point-by-point responses are shown as follows:

1. The Introduction Section was well designed, and the lecture easily flows toward the objectives presented at its end. However, only one article cited in this section was published in the last five years and nine from the 23 of them were published in the last century. This selection makes us wonder the relevance of the research object and the addition that the results to be obtained can bring to what is already described in the literature.

Response: Thanks for the constructive comments. The authors have searched and added more suitable published studies as references in the Introduction Section of the revised version of the paper. Please see the following lists of the additional articles cited in this revised manuscript. While searching for these updated citations, the authors also found that the research works related to children’s handwriting kinetics have become more scarce in recent years. Therefore, in addition to classic articles related to handwriting kinetics published several decades ago, most of the studies, as the reviewer mentioned, are works from the period 2005 to 2015, which was an excellent time period for handwriting researchers to discuss the pen tip force based on the mature writing tablet (Wacom) system. However, direct evidence regarding the finger/digit forces acting on the pen barrel is still scant, so our team (and other previous works cited in this manuscript) established a Digit Forces Acquisition pen that is similar to a real pen (either shape or weight/size). With technology that can measure small forces incorporating pressure sensors as well as the 3D printing technical advancements, we believe that handwriting kinetics-related studies may be conducted easily for further the purpose of discussing the mechanisms of handwriting motor or processing functions. Thanks again for the suggestion, which motived us to updated our citations in the revised Introduction section.

Hochhauser M, Wagner M, Shvalb N. Assessment of children's writing features: A pilot method study of pen-grip kinetics and writing surface pressure. Assist Technol. 2021. Epub 2021/07/22. doi: 10.1080/10400435.2021.1956640.

Lin Y-C, Chao Y-L, Hsu C-H, Hsu H-M, Chen P-T, Kuo L-C. The effect of task complexity on handwriting kinetics. Canadian Journal of Occupational Therapy. 2019;86(2):158-68. doi: 10.1177/0008417419832327

Gatouillat A, Dumortier A, Perera S, Badr Y, Gehin C, Sejdic E. Analysis of the pen pressure and grip force signal during basic drawing tasks: The timing and speed changes impact drawing characteristics. Comput Biol Med. 2017;87:124-31. Epub 2017/06/06. doi: 10.1016/j.compbiomed.2017.05.020.

2. This wondering rises when the sample is detailed. Three HD forms are found in the sample: “a cognitive learning dysfunction subtype (25%), a motor impairment subtype (28.6%), and a severe hybrid subtype (46.4%).” Further on, the text describes each of these types of HD: “The severe hybrid subtype is the most serious subtype, reflecting severe deficits in all five dimensions and poor perceptual motor ability” ... “The motor impairment subtype is mainly characterized with deficits in writing speed, construction of characters, DCD, writing automation, and fine motor control” … “Children classified into the cognitive learning dysfunction subtype lack appropriate, efficient learning strategies”. In this sense, the differences between each subtype are significant, either by severity or specificity. However, the sample is treated as a single group so that there can be statistical strength. 

Response: Thanks for the comments. Considering that the original and primary purpose of this study was to reveal the difference in pen grip kinetics in the handwriting performance of an HD group and a control group, the CHEF was used in this study mainly to verify that the children in the HD group indeed had handwriting problems. Distinguishing among the differences in the writing kinetics among the HD subtypes, however, was not the original purpose of this study. Nevertheless, the authors agree with the reviewer’s comment that the differences in the writing characteristics among each HD subtype might be significant or interesting topics to be explored further. Therefore, the authors have stated this in the Discussion section and the Limitations and Future Studies part of this revised version as “However, the data in this study were analyzed in children with uneven percentages of distribution among all three subtypes of HD. Since HD is very complex, and an assessment for HD classification has only been developed recently, the results obtained in this study should be interpreted with caution when generalizing the findings to children with different types of HD” and ” Finally, although children with three subtypes of HD were reported according to the CHEF classification, this study did not analyze the differences of the characteristics of writing kinetics among the different HD subtypes due to the small sample size and uneven percentages of distribution among the subtypes in the current study sample. A future study with a larger sampling scale is suggested to compare the differences of the writing features among HD subtypes.”

Please refer to Page 16 lines 262-266 and Page 21 lines 378-383, respectively.

3. Finally, at the beginning of the discussion, when commenting on the homogeneity of manual dexterity between the groups, it is stated that “this might be related to the fact that most of the writing difficulty subtypes were “motor impairment” and “severe hybrid,” which are both related to poor motor performance during handwriting”, which demonstrates an important reductionism of the type "severe hybrid". This statement also puts a quarter of the sample out of the discussion by assuming that the argument used for both of that types would not apply to the "cognitive learning dysfunction" type.

Response: Thanks for the comments. The authors agree with the reviewer’s viewpoint. The original statement was misused to assume that the argument used for both of that types would not apply to the "cognitive learning dysfunction" type. The authors thus deleted the original sentences “This might be related to …poor motor performance during handwriting.” and added the following sentences: “However, the data in this study were analyzed in children with uneven percentages of distribution among all three subtypes of HD. Since HD is very complex, and an assessment for HD classification has only been developed recently, the results obtained in this study should be interpreted with caution when generalizing the findings to children with different types of HD.” at the end of this paragraph. As mentioned in the previous question (point 2), the characteristics of each HD subtype could be an important issue. Therefore, the authors have also stated this issue as one of the study limitations in the Limitations and Future Studies section.

Please refer to Page 16 lines 262-266 and Page 21 lines 375-376, respectively.

4. In addition, the study observed five different categories of handwriting grip with different intra-group and – mainly – between-groups distributions. There is no mention of the distribution of these categories of handwriting grip in the three types of HD in the sample.

Response: Thanks for the comments. There are two reasons why the authors did not mention the distribution of these handwriting grip categories in the three types of HD in our recruited participants. The first reason is that all the children recruited in this study could use the tripod grasp to write. This grip type was recorded as the main grip type when a child uses a regular or typical pencil to write because it is usually taught as the standard writing grip pattern by the school teachers in our education system. That is the reason all the recruited children could use the force acquisition pen (FAP) to write during the assigned tasks. According to our design rationale (to fit the size and weight of a regular pen) for the FAP, only three sensors could be installed on the pen barrel, so only a tripod pen grasp was allowed in our handwriting testing. The other reason is that all four grip types for the HD group among our recruited children were the so-called mature grip type according to the previous classifications (shown in the following lists). Since all the participants in the HD group had already become used to the mature grip type in their writing performance, the authors thus decided not to specifically mention the details of the grip patterns in the text. However, because our original statement was not clear, the authors added one sentence ”Although all the children had already been taught to use the dynamic tripod grasp to write in school, the most preferred and frequent manner in which they gripped the pencil was recorded.” to clarify the method for defining the grip types in this study. The following table also shows the distribution of these categories of handwriting grip in the three types of HD among the recruited children.

In addition, in the design of the FAP, we had to constrain the pen-grip pattern to the tripod grip type for everyone, which may have limited a full understanding of grip kinetics among different pen grip patterns. Therefore, the authors also added this issue as one of the study limitations in the Limitations and Future Studies part as “Further, a force acquisition pen that is capable for detecting grip kinetics from different pen grip patterns other than the tripod grasp is still needed.”.

Please refer to Page 7 lines 128-129 and Page 21 lines 375-376.

HD subtype Grip pattern Numbers of subjects (%)

Cognitive learning dysfunction Dynamic tripod 4 (57.1)

 Lateral tripod 2 (28.6)

 Quadruped 1 (14.3)

 Total 7 (100)

Motor impairment Lateral tripod 6 (75)

 Lateral quadruped 2 (25)

 Total 8 (100)

Severe hybrid Dynamic tripod 1 (7.7)

 Lateral tripod 6 (46.2)

 Quadruped 2 (15.4)

 Lateral quadruped 4 (30.8)

 Total 13 (100)

Mature grips citations:

Koziatek, S. M., & Powell, N. J. (2003). Pencil grips, legibility, and speed of fourth-graders’ writing in cursive. American Journal of Occupational Therapy, 57(3), 284-288.

Tseng, M. H. (1998). Development of pencil grip position in preschool children. The Occupational Therapy Journal of Research, 18(4), 207-224.

5. There is an important phrase related to this topic whose wording is ambiguous in my opinion:

“Although the positions of the three force sensors on FAP are adjustable, in this study, all participants used the same position setting for the force sensors based on their own grip.” What does it mean? That all children used the same position setting of the sensors in the pen, regardless of how each child grasped it? Or that each child had an individualized configuration of sensors according to their form of grasping? Or, yet, that there were several configurations, one for each form of handwriting grasping, but not one for each child according to their natural positioning?

This questioning is important because it can indicate whether there was an extra constraint factor for the HD group. If the positioning was unique, its effects on the HD group may have been greater than on the group without HD. If the positioning was determined by the way of grasping, how many subgroups were formed? And if the positioning was individualized by a child, what did you mean by "same position setting"? Therefore, I believe that, by not adequately considering the heterogeneity of the HD sample, the authors compromised the reliability of the results. As stated by them, “due to the small sample size and uneven percentages among the subtypes, differences between HD subtypes were not analyzed”.

Response: Thanks for the comments. The sentence “Although the positions of the three force sensors on FAP are adjustable, in this study, all participants used the same position setting for the force sensors based on their own grip.” in our original text may have been too vague. To improve readability, the authors have rephrased the statement as follows: “Although the positions of the three force sensors on the FAP were adjustable, no participant in this study, however, requested the examiner to adjust the position of the sensors due to discomfort or an awkward hand posture.” in the Procedures part of this revised version. Please refer to Page 9 lines 181-182, and Page 10 line 183.

Again, the authors agree with the reviewer’s viewpoint regarding the heterogeneity of the HD participants. Unfortunately, as in our previous response, the objective of this was to reveal the differences in pen grip kinetics in handwriting performance for the HD group and control group. Although distinguishing the differences in the writing kinetics among the HD subtypes is also a critical issue to be investigated, this was not the focus of the present study. Therefore, the authors stated this issue in the Discussion section and the Limitations and Future Studies part of this revised version as follows: “However, the data in this study were analyzed in children with uneven percentages of distribution among all three subtypes of HD. Since HD is very complex, and an assessment for HD classification has only been developed recently, the results obtained in this study should be interpreted with caution when generalizing the findings to children with different types of HD.” and “ Finally, although children with three subtypes of HD were reported according to the CHEF classification, this study did not analyze the differences of the characteristics of writing kinetics among the different HD subtypes due to the small sample size and uneven percentages of distribution among the subtypes in the current study sample. A future study with a larger sampling scale is suggested to compare the differences of the writing features among HD subtypes.” Please refer to Page 16 lines 262-266 and Page 21 lines 378-383, respectively.

6. Some minor considerations:

The model of direct citations used in the article (citing only the first two authors) is not adequate, since it gives a single comma the power to indicate that there are other authors besides the two mentioned. Forms such as "Author and colleagues", "Author and collaborators" and even "Author et al." are preferable in these cases.

Response: Thanks for the reviewer’s correction. The authors have revised the description as follows: “However, Chau and colleagues [15] showed different results in their work regarding pen-grip activity.”, “Lin et al. [13] enrolled 181 children ranging in age from 5 to 12 years old and found that the younger children exhibited more force variations and lower frequency of adjustment in the amount of force applied from each digit.”, “Regarding the grip-to-tip ratio, which is the total force applied from the grip divided by the force applied from the pen-tip, Lin et al. [13] found that older children tend to apply force on the pen barrel rather than pushing the pen-tip downward. Similarly, Chau et al. [15] found that compared to children with CP, children with typical development generate higher grip-to-tip ratios.”, “Ghali et al. [22] suggested that each person has a specific, recognizable force distribution on the pen barrel while writing. Shim et al. [23] also proposed a system measuring contact forces from three digits.”, “Drotár et al. [35] suggested that in-air movement during handwriting may involve cognitive processes such as motor planning and programming of motor sequences.”, and “Rosenblum et al. [16] found that children with poor handwriting had a lower in-air to total time ratio and a lower speed.”.

Please refer to Page 4 line 66-67; Page 5 lines 78-80; Page 5 lines 83-87; Page 5 lines 89-91; Page 12 lines 202-203; Page 17 lines 282-283.

7. Although it could be considered as implied, the text does not formally indicate whether the sample is composed only by right-handed children and does not comment on the consistency of manuality. They belong to a key age range regarding manuality definition, and the topic deserves to be considered in the characterization of the sample (Scharoun & Bryden, 2014). Scharoun, S. M. and Bryden, P. J. Hand preference, performance abilities, and hand selection in children. Front Psychol. 2014; 5: 82. doi: 10.3389/fpsyg.2014.00082

Response: Thanks for the reviewer’s comment and suggestion. There were three left-handed subjects in the HD group. The authors revised the description as follows: “Thirty-six children without HD (Control group, 18 girls and 18 boys, mean age 7.97±0.57 years, all right-handed) …3 left-handed and 25 right-handed)… therapy practitioners.”.

Please refer to Page 8 line 111-113 and Page 9 line 114-116.

8. The Cronbach’s alpha value for the BOT-2 test is 0.92; however, ideal values for Cronbach’s alpha range from 0.7 to 0.9. Values greater than 0.9 indicate test redundancy (Streiner, 2003). Streiner, D. L. Being inconsistent about consistency: when coefficient alpha does and doesn’t matter. Journal of Personality Assessment. 2003; 80:217-222. doi: 10.1207/S15327752JPA8003_01

Response: Thanks for the comments. The authors agree with the reviewer’s viewpoint. There are two fine motor skill assessment tools commonly used in hospitals for evaluating children’s fine motor capability in Taiwan. One is the BOT-2, and the other one is the Movement ABC (Movement Assessment Battery for Children). Both tests are widely used in scientific studies. The BOT-2 uses the same tests for age from 4 to 21 years old. However, the content of the tests in the Movement ABC should be changed depending on the age of the subjects. In order to make further comparisons with other similar studies in the field convenient, the BOT-2 was chosen for our experiment. Although the Cronbach’s alpha value for the BOT-2 test indicates test redundancy, it seems to be one of the best choices to provide an objective assessment of children’s fine motor functions in a clinical setting, as compared to the Movement ABC.

Croce, R. V., Horvat, M., & McCarthy, E. (2001). Reliability and concurrent validity of the movement assessment battery for children. Perceptual and motor skills, 93(1), 275-280.

9. Right following the phrase “There have been few studies reporting on applied force while pen-lifting” there are no references to these few studies, nor a discussion comparing the results obtained in the study with this literature, which is very relevant here.

Response: Thanks for the reviewer’s correction. The authors revised the statements as follows: “To the best of our knowledge, there are no studies in the literature on applied digit force during pen-lifting. This study may thus be the first to report these data based on the handwriting performance of children.”. 

Please refer to Page 17 lines 276 to 278.

 

Reviewer #2: 

The aim of this study was to characterise the pen-grip kinetics profile of children with and without handwriting difficulties (HD), an issue that has received little attention to date. Despite the relevance of the topic, certain aspects limit the impact of the study, for example, the absence of an in-depth analysis of past studies; the use of a heterogeneous sample of children with handwriting difficulties; and the statistical approach to data analysis. The points that need to be clarified and strengthened are detailed below.

We sincerely appreciate the Reviewer’s thorough review and valuable suggestions. The authors revised the manuscript according to your constructive comments. The point-by-point responses are shown as follows:

Introduction

1. The introduction requires a more detailed review of previous studies. According to the authors, “the results of previous studies related to children’s handwriting kinetics remained inconsistent” (p.7, lines 2-3), meaning that one of the main motivations of the present study was to provide more evidence in regard to the issue. According to the above, the reader would expect to be provided with possible explanations for the heterogeneity of previous results. 

However, only in the case of findings concerning variations in terms of grip force were certain methodological reasons mentioned as a source of inconsistency (p.6, lines 13-15). A more in-depth analysis of previous findings would make for a more compelling argument as to the specific contribution of the present study.

Response: Thanks for the reviewer’s constructive comments and corrections. The authors agree with the reviewer’s viewpoints regarding the flaws of the description about “the inconsistent issue” in our original Introduction section. In order to concentrate on the main focus of this study, which was an attempt to present “the lack of previous works regarding comprehension of digit forces acting on a pen barrel during handwriting” and also avoid misleading the readers as to the themes addressed in this study, the authors removed the irrelevant statement “the results of previous studies related to children’s handwriting kinetics remain inconsistent, and…” in the Introduction section of the revised version of the paper. According to the context of this paragraph, we rephrased the sentence as “Since studies on digit force are still limited, this study was designed as an observational study to provide insight into the issue.” so that inconsistencies in the handwriting kinetics (especially those related to the pen-tip force) will not be the main argument in this study. Please refer to Page 6 lines 93-94.

In addition, the authors have searched and added more proper published works as the references in the revised Introduction section: Please refer to Page 6 line 93. While updating this section, the authors also found that research works related to children’s handwriting kinetics have become scarce in recent years. Besides those classic articles related to handwriting kinetics published several decades ago, most of the studies were the works around 2005 to 2015 which were the fabulous time period for the handwriting researchers to discussed the pen tip force based on the mature tablet (Wacom) system. However, the comprehension of the direct evidence regarding the finger/digit forces acting on the pen barrel was still scant so that our team (and other previous works which had been cited in this manuscript) had established a digit forces acquisition pen that was similar to the real pen (either shape or weight/size). With the technology of conducting small force or pressure sensors as well as the 3D printing technical advancements, we believe that handwriting kinetics-related studies might be conducted easily for further discussing the mechanisms of handwriting motor or processing functions. Thanks again for the reviewer’s suggestions and corrections which motived us to update our citations and improve the readability of the revised version of the paper.

2. The research questions overlap to a certain degree. It would be advisable to reformulate them, as this would also help to improve the statistical analysis strategy.

Response: Thanks for the reviewer’s constructive suggestions. The authors revised the statements of research questions as “(1) Are there any differences in the fine motor skills and pen-grip kinetics between children with HD and without HD, and (2) are there any role differences among the thumb, index, and middle finger?”. Please refer to Page 6 lines 99-101. In addition, the statistical analyses used for examining the questions were also changed to the use of ANCOVAs. The description was revised as “To address research question one, an analysis of covariance (ANCOVA) controlling for age was used to analyze the experimental data to determine the between-group differences in the parameters and raw BOT-2 scores. To address research question two, since the participants were compared in two groups, a mixed-model ANCOVA including one between factor (group: with HD vs. without HD) and one within factor (role of the digits: thumb vs. index finger vs. middle finger) was performed on the AFwhole task, the CVFwhole task, and the NFFPS whole task, with age as a covariate. Bonferroni tests were used as post-hoc tests.” Please refer to Page 12 lines 214-219; Page13 lines 220-221. 

Method

3. My main concerns in this section are in regard to the sample. The group of children classified as having HD is highly heterogeneous according to the profiles extracted from the Chinese Handwriting Evaluation Form (CHEF), so it is difficult to understand the implications of the differences found later with regard to the control group. The difference in the age range between the groups should also be emphasised, as the children with HD had almost an additional year of handwriting practice compared to their non-HD peers, a fact which could explain the lack of differences in some measures. 

Response: Thanks for the comments. The authors agree with the comment regarding the heterogeneity of children in the HD group. Considering the original and main purpose of this study was to reveal the difference in pen grip kinetics in handwriting performance between an HD group and a control group, the details of the characteristics of the children in HD group were not specifically analyzed. However, since there are very few studies that discuss the digit force in children with HD, the authors believe this study provides new evidence and concept as a pilot study. Again, the authors agree with the reviewer’s concerns. Therefore, the authors stated this issue in the Discussion section and the Limitations and Future Studies part of this revised version as follows: “However, the data in this study were analyzed in children with uneven percentages of distribution among all three subtypes of HD. Since HD is very complex, and an assessment for HD classification has only been developed recently, the results obtained in this study should be interpreted with caution when generalizing the findings to children with different types of HD.” and ” Finally, although children with three subtypes of HD were reported according to the CHEF classification, this study did not analyze the differences of the characteristics of writing kinetics among the different HD subtypes due to the small sample size and uneven percentages of distribution among the subtypes in the current study sample. A future study with a larger sampling scale is suggested to compare the differences of the writing features among HD subtypes.” Please refer to Page 16 lines 262-266 and Page 21 lines 378-383, respectively.

4. The pen-grip patterns were not distributed in the same way between the groups with and without HD. The lateral quadruped grasp was more prevalent within the HD group, while the dynamic tripod grasp was more prevalent within the control group. Do you think this could be a source of explanation for the differences found between the groups? This point is especially relevant if we consider that, as I understand it, the positions of the three force sensors were fixed with a tripod grasp pattern: “Although the positions of the three force sensors on FAP are adjustable, in this study, all participants used the same position setting for the force sensors based on their own grip. They were requested to trace numbers in each trial, with three trials each. They had a one-minute rest interval between each trial and enough time to practice using the FAP with a tripod grasp before the formal trials” (p.10, lines 18-22). 

Response: Thanks for the comments. The authors agree with the comments that the pen-grip patterns could provide an explanation for the between-group differences. While looking carefully at the distribution of pen-grip patterns between groups, the difference in the dynamic tripod was about 8%, and in the lateral quadpod, it was about 11%. However, considering the small difference and the small sample size, the issue of the distribution of these categories of handwriting grip in the two groups was not discussed. Again, the authors deeply appreciate the reviewer’s constructive comments and understand this issue is worthy of further investigation. The authors thus added the statement regarding the influence of pen-grip type into the study limitations in the Limitations and Future Studies part as “Further, a force acquisition pen that is capable for detecting grip kinetics from different pen grip patterns other than the tripod grasp is still needed.” Please refer to Page 21 lines 375-376.

Results

5. This section requires a major revision.

The analysis strategy used to answer the two questions could be improved. To answer research question one (Are there any differences in the fine motor skills and pen-grip kinetics between children with HD and without HD?), a series of t-tests were used to compare the groups using variables that are correlated; this is not a good idea given the increased chance of a type I error. The differences in age should have been controlled.

Response: Thanks for the comments and suggestions. Previous studies (see the following lists) usually examined the handwriting performance with different parameters in different periods or under different conditions to determine the specific writing control pattern. Although the variables were correlated, each variable could represent its specific control patterns and writing features. Therefore, the authors used these variables to present their specific control patterns and clinical meanings in Table 1.

Since the age was not controlled well, the authors revised the discussion as follows: “The mean age of children with HD was more than 8 months older than the age of the children without HD. This indicated that the children with HD had almost an additional year of handwriting practice compared to their non-HD peers in this study.” to emphasize the age difference between groups. In addition, as mentioned in the previous question (point 2.), the authors now use ANCOVAs to control for age to address both research questions.

Please refer to Page 7 lines 116-118.

Citations:

Rosenblum, S., & Livneh-Zirinski, M. (2008). Handwriting process and product characteristics of children diagnosed with developmental coordination disorder. Human movement science, 27(2), 200-214.

Chang, S. H., & Yu, N. Y. (2013). Handwriting movement analyses comparing first and second graders with normal or dysgraphic characteristics. Research in developmental disabilities, 34(9), 2433-2441.

Horie, S., & Shibata, K. (2018). Quantitative evaluation of handwriting: factors that affect pen operating skills. Journal of physical therapy science, 30(8), 971-975.

6. To answer research question two (Are there any differences in pen-grip kinetics among different digits that indicate corresponding roles?), three one-way ANOVAs were used for each group. At this point, one wonders why the authors were not interested in the differences between the groups, although this is related to the problem of overlap in the research questions. As the comparison between groups was not of interest to the authors, they compared the force grip of the three digits for each group and parameter. A less complex and more robust approach could have been achieved using a mixed model.

Response: Thanks for the comments and suggestions. The authors revised the statements related to the research questions to clarify the study questions of interest. In addition, owing to the clarification of our research questions, the authors also revised the statistical analysis for this second research question from a one-way ANOVA to a mixed-model ANCOVA. This is explained in the paper as follows: “To address research question one, an analysis of covariance (ANCOVA) controlling for age was used to analyze the experimental data to determine the between-group differences in the parameters and raw BOT-2 scores. To address research question two, since the participants were compared in two groups, a mixed-model ANCOVA including one between factor (group: with HD vs. without HD) and one within factor (role of the digits: thumb vs. index finger vs. middle finger) was performed on the AFwhole task, the CVFwhole task, and the NFFPS whole task, with age as a covariate. Bonferroni tests were used as post-hoc tests.” The Results and Discussion sections related to the role differences in the digits have also been revised accordingly. 

Please refer to Page 12 lines 214-219 and Page 13 lines 220-221; Page 19 lines 333-345 and Page 20 lines 346-361.

Discussion

7. Given that the main argument of the study was based on the inconsistencies in the findings of previous studies, I would have expected a discussion reporting explicitly on how the results of the present work help to clarify these inconsistencies. However, the discussion does not address this. 

The authors appropriately highlight the limitations of the work, but do not address the practical implications that the results could have for the identification and intervention of children with HD.

Response: Thanks for the comments and suggestions. As the authors mentioned in their answer to question #1, the main argument in the revised version was changed. The inconsistency in the handwriting kinetics (pen-tip force) is not the main argument in this study. The main focus is more on ‘the lack of studies regarding digit force.’ Therefore, the statement reporting on how the results of the present work might help to clarify these inconsistencies is not explicitly addressed in the Discussion section.

Since the recruited group of HD children could have been heterogeneous in this study, this may lead to it being difficult to directly generalize the findings to all children with HD. This is why the authors have carefully concluded and tried not to overexplain the findings of this study. The implications are mentioned in the Conclusion section as follows: “On the basis of the results, our findings suggested that when intervening in children’s handwriting problems, it should be noted if the children loosen the gripping digits even when the digits appear to be touching the pen barrel during writing, especially at the moment when they lift the pen to write the next stroke. Digit role differences could also be considered when evaluating writing. Compared to the children with HD, the children without HD used a more static method by which to apply the force required to grip the pen and generated faster force adjustments to manipulate the pen from a kinetic point of view.”. Please refer to Page 21 lines 386-391 and Page 22 lines 392-394. 

Since there are very few studies discussing digit force in children with HD, the authors believe this study provides new evidence and concept as a pilot study. This study was an attempt to explore the digit force quantitatively in children with HD via the use of a force acquisition pen. Although the limitations, such as the heterogeneity of the sample, applicability of the FAP, and the comparison of grip-types exist, this study still provides evidence on the role of force control among digits and a pen during a writing task. Furthermore, the HD classification is still a critical issue to be further explored. As for the CHEF, which is the only published and purchasable standard tool in the traditional Chinese handwriting system, the limitations of this tool, such as the small number of published studies regarding its use or the fact that it is a questionnaire rather a pencil paper test, indicate a need for more research evidence and in-depth discussion. Once the relationship between digits and a pen are revealed for a specific handwriting style or for a specific handwriting problem, standard principles for digit force control patterns can be constructed. Therefore, this study has shown the difference in the digit kinetics between children with and without HD. In future studies, the development of an understanding of different HD subtypes as well as grip patterns during handwriting tasks should be explored.

---

## [Decision Letter · Decision Letter 1]

5 Jan 2022

PONE-D-21-12263R1Pen-grip Kinetics in Children with and without Handwriting DifficultiesPLOS ONE

Dear Dr. Kuo,

Thank you for submitting your manuscript to PLOS ONE. After careful consideration, we feel that it has merit but does not fully meet PLOS ONE’s publication criteria as it currently stands. Therefore, we invite you to submit a revised version of the manuscript that addresses the points raised during the review process.

We look forward to receiving your revised manuscript.

Kind regards,

Victor Frak, MD, Ph.D

Academic Editor

PLOS ONE

Reviewers' comments:

Reviewer's Responses to Questions

**Comments to the Author**

1. If the authors have adequately addressed your comments raised in a previous round of review and you feel that this manuscript is now acceptable for publication, you may indicate that here to bypass the “Comments to the Author” section, enter your conflict of interest statement in the “Confidential to Editor” section, and submit your "Accept" recommendation.

Reviewer #1: All comments have been addressed

Reviewer #2: (No Response)

2. Is the manuscript technically sound, and do the data support the conclusions?

Reviewer #1: Yes

Reviewer #2: Partly

3. Has the statistical analysis been performed appropriately and rigorously? 

Reviewer #1: Yes

Reviewer #2: No

4. Have the authors made all data underlying the findings in their manuscript fully available?

Reviewer #1: Yes

Reviewer #2: Yes

5. Is the manuscript presented in an intelligible fashion and written in standard English?

Reviewer #1: Yes

Reviewer #2: Yes

6. Review Comments to the Author

Reviewer #1: The authors did a good job of reviewing the proposed article following the 1st round of revision. The changes regarding the bibliography used and the statistical analysis raised the quality of the work and the importance of the results presented here.

Since the control group had 25 children, I believe the HD group would be a little less heterogeneous if the three left-handed children were removed from the sample and from the statistics. That leaves two groups of 25 children, all of them right-handed. However, since the data set are provided with the article, I do not believe that the presence of these children in the sample invalidates the results obtained. It is up to the authors and the editor to assess whether this change is necessary.

The discussion benefited from the changes made to the statistics. However, the three articles added to the Introduction could certainly have been taken up in the discussion, since they are recent and relevant to the discussion of this work.

Reviewer #2: The authors have adequately addressed earlier comments about the introduction section. They have also adjusted the research questions and included “age” as a covariate in the analyses, as suggested. However, limitations remain that have not been properly addressed.

Results section

• With regard to the split-plot analysis for AF, why was it used for the “whole task” condition but not for the “in air” and “on paper” conditions?

• If I have understood correctly, the Group x Digit type interaction is explained by dividing the interaction into two main effects: “Therefore, the simple main effect was then analyzed with ANCOVA (for the group) and One-Way ANOVA (for the digit type)” (lines 244-245). This approach breaks with the logic of an interaction analysis. It needs to be explained by making (orthogonal) contrasts between the levels of the different factors (e.g., differences between HD and control in level 1 [thumb] and 2 [index] vs level 3 [middle]). This may have been done, but the statistical effects associated with the contrasts are not described.

• No effect size is described in any of the analyses.

• The analyses generate redundant results. If CVF, NFFPS and AF (whole task condition) are included in the mixed design, they should not be included in the first analysis.

Another two issues remain unclear:

• The first concerns the heterogeneity of the HD group. The authors include this aspect as a limitation, but it would be necessary to show the descriptive statistics of the study variables for each subtype, perhaps in supplementary material. This would help to explain how the variables work in each subtype and could provide a better understanding of the differences between HD and control.

• The second is related to the procedure, specifically, to the fact that the positions of the three force sensors were fixed with a tripod grasp pattern. The authors state: “Although the positions of the three force sensors on the FAP were adjustable, no participant in this study, however, requested the examiner to adjust the position of the sensors due to discomfort or an awkward hand posture” (lines 181-183). However, they do not indicate whether within the task instructions participants were told that if they are uncomfortable, the position can be adjusted.

In the discussion section, a number of points need to be addressed in depth. For example, the results concerning AF-in-air and AF-in-paper are not interpreted. What are the implications of the fact that the differences between the groups in AF are found in the “in air” condition but not “on paper”?

7. PLOS authors have the option to publish the peer review history of their article (what does this mean?). If published, this will include your full peer review and any attached files.

Reviewer #1: **Yes: **Ronaldo Luis da Silva

Reviewer #2: No

---

## [Author Response · Author response to Decision Letter 1]

18 Feb 2022

Reviewer #1:

The authors did a good job of reviewing the proposed article following the 1st round of revision. The changes regarding the bibliography used and the statistical analysis raised the quality of the work and the importance of the results presented here.

Since the control group had 25 children, I believe the HD group would be a little less heterogeneous if the three left-handed children were removed from the sample and from the statistics. 

That leaves two groups of 25 children, all of them right-handed. However, since the data set are provided with the article, I do not believe that the presence of these children in the sample invalidates the results obtained. It is up to the authors and the editor to assess whether this change is necessary.

The discussion benefited from the changes made to the statistics. However, the three articles added to the Introduction could certainly have been taken up in the discussion, since they are recent and relevant to the discussion of this work.

Response: We sincerely appreciate the Reviewer’s thorough review and valuable suggestions. The authors have done a sensitivity analysis by removing three left-handed children. The result is similar to the original data without removing left-handed children. In addition, we have consulted the Biostatistics Consulting Center, Clinical Medicine Research Center, National Cheng Kung University Hospital for the rationality and applicability of the statistics used in this study. For the statistics results after removing three left-handed children, please refer to the following Table A and Table B.

Table A. ANCOVA summary table for determining the effect of the digit, group, and their interaction— removed 3 left-handed children

Kinetics of handwriting Covariance

(Age) Group Digit Group � Digit 

 F Sig. ηp2 F Sig. ηp2 F Sig. ηp2 F Sig. ηp2 Post-hoc

AFon paper 0.55 .46 .01 4.87 .03* .08 2.57 .08 .04 0.44 .65 .01 Ct>HD

AFin air 0.53 .47 .01 12.74 .00** .18 1.63 .20 .03 1.08 .34 .02 Ct>HD

AFwhole task 0.69 .41 .01 5.94 .02* .09 2.18 .12 .04 0.91 .40 .02 Ct>HD

CVFwhole task 2.02 .16 .03 6.75 .01 .10 1.36 .26 .02 8.94 .00** .13 Interactiona

NFFPSwhole task 0.27 .61 .01 5.73 .02* .09 1.60 .21 .03 1.46 .24 .03 Ct>HD

Note. AF, average force; CVF, coefficient of variation in force; NFFPS, number of force fluctuations per second; Ct, control group; HD, handwriting difficulty group. Post-hoc type: Bonferroni, *p < 0.05 ,**p < 0.01

a Simple main effect:

ANCOVA for group difference between digitse(three levles):

level 1[Thumb], F=1.06, p=.31, ηp2 =.018; level 2[Index], F=20.73, p=.00, ηp2 =.263; level 3[Middle], F=.09, p=.77, ηp2 =.001;

One-way ANOVA for digit type difference between groups: 

[HD], F=8.18, p =.00, ηp2 =.254; Post-hoc: Thumb vs Index, p =.01; Thumb vs Middle, p =1.00; Index vs Middle, p =.02

[Ct], F=5.29, p =.02, ηp2 =.131; Post-hoc: Thumb vs Index, p =.02; Thumb vs Middle, p =0.43; Index vs Middle, p =.03

Table B. Descriptive statistics parameters in two groups and the ANCOVA result for between group difference after removing 3 left-handed children

Kinetics of handwriting

(sources of force) Group 

 Control (n=36) HD (n=25) 

 Mean SD Mean SD Sig.a ηp2 a

Fine motor skills 

 Fine motor precision 38.44 2.16 33.20 5.13 .000** .484

 Fine motor integration 37.22 2.04 33.96 5.74 .000** .237

 Manual dexterity 25.94 2.91 25.80 5.19 .141 .037

Average Force (AF), unit: Newton 

 Thumb -on paper 3.07 1.31 2.62 0.61 

 Thumb -in air 1.56 0.87 1.08 0.30 

 Thumb -whole task 2.70 1.22 2.21 0.53 

 Index -on paper 2.11 0.77 1.68 0.62 

 Index -in air 1.15 0.55 0.63 0.22 

 Index -whole task 1.87 0.72 1.42 0.47 

 Middle -on paper 1.34 0.75 1.13 0.57 

 Middle -in air 1.05 0.49 0.77 0.32 

 Middle -whole task 1.29 0.69 1.11 0.54 

 Pen-Tip -on paper 0.95 0.29 0.75 0.33 .012* .105

Task Time (TT) 

 TT -on paper (ratio) 0.71 0.08 0.72 0.05 .672 .003

 TT -in air (ratio) 0.29 0.08 0.28 0.05 .708 .002

 TT -whole task (s) 23.46 7.49 27.29 10.61 .046* .067

Coefficient of Variation in Force (CVF) 

 Thumb -whole task 0.38 0.09 0.40 0.08 

 Index -whole task 0.35 0.08 0.49 0.15 

 Middle -whole task 0.42 0.14 0.41 0.12 

 Pen-Tip -on paper 0.36 0.05 0.34 0.07 .245 .023

Number of Force Fluctuations per second (NFFPS) 

 Thumb -whole task 1.34 0.24 1.25 0.17 

 Index -whole task 1.36 0.17 1.26 0.23 

 Middle -whole task 1.59 0.38 1.39 0.27 

Force Ratio (FR) -on paper 0.16 0.05 0.10 0.04 .000** .229

Note. SD = standard deviation; HD = handwriting difficulty; *p < 0.05,**p < 0.01

a Results of the ANCOVA correcting for age as a covariate

 

Reviewer #2:

The authors have adequately addressed earlier comments about the introduction section. They have also adjusted the research questions and included “age” as a covariate in the analyses, as suggested. However, limitations remain that have not been properly addressed.

Response: We sincerely appreciate the Reviewer’s thorough review and valuable suggestions. The authors revised the manuscript according to your constructive comments. In addition, we have consulted the Biostatistics Consulting Center, Clinical Medicine Research Center, National Cheng Kung University Hospital for the rationality and applicability of the statistics used in this study. The point-by-point responses are shown as follows:

1. Results section

With regard to the split-plot analysis for AF, why was it used for the “whole task” condition but not for the “in air” and “on paper” conditions?

Response: Thanks for the Reviewer’s constructive comments and correction.

The authors overlooked the analysis for AF for “in air” and “on paper” conditions for research question 2. Since the condition may represent different control or writing patterns during handwriting, it should be analyzed in both research questions. Thanks to the Reviewer’s correction again. The authors added these two conditions into the analysis and revised the manuscript and also Table 3. The description was revised as “A mixed-model ANCOVA including one between factor (group: with HD vs. without HD) and one within factor (role of the digits: thumb vs. index finger vs. middle finger) was performed on the AFon paper, the AFin air, the AFwhole task, the CVFwhole task, and the NFFPS whole task, with age as a covariate.”; “The results of the mixed-model ANCOVA showed significant differences in the AFon paper, the AFin air, and the AFwhole task of the three digits. All these AFs were significantly lower in the HD group.”; “Table 3 shows the results for the effects of group and digit type on the AFin air, AFon paper, AFwhole task, CVFwhole task, and NFFPSwhole task. There was no interaction effect between the group and digit type when analyzing the AFin air, AFon paper, AFwhole task and NFFPSwhole task via the mixed-model ANCOVA controlling for age. No significant difference among digit types was found on the AFin air, AFon paper, and AFwhole task or NFFPSwhole task. The scores for the AFin air, AFon paper, AFwhole task and NFFPSwhole task were found to be significantly greater in the control group.”.

Please refer to Page 12 lines 215-220; Page 13 line 221; Page 13 lines 229-231; Page 15 lines 242-247; Page 16 Table 3.

2. If I have understood correctly, the Group x Digit type interaction is explained by dividing the interaction into two main effects: “Therefore, the simple main effect was then analyzed with ANCOVA (for the group) and One-Way ANOVA (for the digit type)” (lines 244-245). This approach breaks with the logic of an interaction analysis. It needs to be explained by making (orthogonal) contrasts between the levels of the different factors (e.g., differences between HD and control in level 1 [thumb] and 2 [index] vs level 3 [middle]). This may have been done, but the statistical effects associated with the contrasts are not described.

Response: Thanks for the Reviewer’s constructive comments and corrections. The authors added the descriptions in the revised manuscript as “The ANCOVA result showed that only the CVFwhole task of the index in the HD group was significantly greater than in the control group (difference between groups in three digit types [Thumb], F=1.13, p=.29, ηp2 =.018; [Index], F=20.30, p=.00, ηp2 =.250; [Middle], F=0.41, p=.53, ηp2 =.007). The One-Way ANOVA result showed that significant differences were found in HD (difference among digits, F=6.61, p =.00, ηp2 =.197) and control (difference among digits, F=5.29, p =.02, ηp2 =.131) group. The post-hoc tests showed that the CVFwhole task of the thumb and middle finger were found to be significantly higher than the CVFwhole task of the index finger in the control group (Thumb vs. Index finger, p =.02; Thumb vs. Middle finger, p =.43; Index finger vs. Middle finger, p =.03), and the CVFwhole task of the index finger was found to be significantly higher than the CVFwhole task of the thumb in the HD group (Thumb vs. Index finger, p =.01; Thumb vs. Middle finger, p =.43; Index finger vs. Middle finger, p =.14).”. Please refer to Page 15 lines 250-261.

3. No effect size is described in any of the analyses.

Response: Thanks for the Reviewer’s constructive suggestion. The authors added the effect size (ηp2) in the analyses. Please refer to Page 14 Table 2 and Page 15 lines 252-255 and Page 16 Table 3.

4. The analyses generate redundant results. If CVF, NFFPS and AF (whole task condition) are included in the mixed design, they should not be included in the first analysis.

Response: Thanks for the Reviewer’s corrections. The authors revised the parameters used in the two different ANCOVA analyses to avoid generating redundant results. The description was revised as “An analysis of covariance (ANCOVA) controlling for age was used to analyze the experimental data to determine the between-group differences in the parameters (the AFon paper of the pen-tip, the TTs, the CVFon paper of the pen-tip, and the FR ) and raw BOT-2 scores. A mixed-model ANCOVA including one between factor (group: with HD vs. without HD) and one within factor (role of the digits: thumb vs. index finger vs. middle finger) was performed on the AFon paper, the AFin air, the AFwhole task, the CVFwhole task, and the NFFPS whole task, with age as a covariate.”. Table 2 and Table 3 were also revised. Please refer to Page 12 lines 215-220, Page 13 line 221; Page 14 Table 2; Page 16 Table 3. 

5. Another two issues remain unclear:

The first concerns the heterogeneity of the HD group. The authors include this aspect as a limitation, but it would be necessary to show the descriptive statistics of the study variables for each subtype, perhaps in supplementary material. This would help to explain how the variables work in each subtype and could provide a better understanding of the differences between HD and control.

Response: Thanks for the Reviewer’s constructive comments. The authors have added a summary table to show the descriptive statistics of the study variables for each subtype in the supplementary material. 

Please refer to the supporting information, S1 Table. Summary Table of the Parameters in Three Subtypes of HD Group.

6. The second is related to the procedure, specifically, to the fact that the positions of the three force sensors were fixed with a tripod grasp pattern. The authors state: “Although the positions of the three force sensors on the FAP were adjustable, no participant in this study, however, requested the examiner to adjust the position of the sensors due to discomfort or an awkward hand posture” (lines 181-183). However, they do not indicate whether within the task instructions participants were told that if they are uncomfortable, the position can be adjusted.

Response: Thanks for the Reviewer’s constructive comments and corrections. Yes, all the participants were told that the position could be adjusted if uncomfortable. However, the original manuscript was not well stated. Therefore, the authors revised the statements as ” Although the positions of the three force sensors on the FAP were adjustable, no participant in this study, however, requested the examiner to adjust the position of the sensors due to discomfort or an awkward hand posture when the examiners asked them if they needed to adjust the position of the sensors.” Please refer to Page 9 lines 181-182, and Page 10 lines 183-184.

7. In the discussion section, a number of points need to be addressed in depth. For example, the results concerning AF-in-air and AF-in-paper are not interpreted. What are the implications of the fact that the differences between the groups in AF are found in the “in air” condition but not “on paper”?

Response: Thanks for the Reviewer’s constructive comments.

The authors revised the manuscript and added new statements as “Based on the result, loosening the fingers when lifting the pen might influence children’s writing performance. When the pen is picked up, the constantly applied force from the digits is not necessary due to not making the ink on paper. It may not be necessary to constantly apply the force in the energy-saving perspective. On the other hand, it may reduce the frequency to enlarge the applied force. Therefore, one possibility is that these children could not generate the force constantly. Another possibility is that these children do not know how to constantly apply force. However, more evidence is needed to confirm this hypothesis, such as the muscle endurance test or skill check test.”, and “Since handwriting is complex, the reason resulting HD is various. Therefore, this study focused mainly on one concept, the writing kinetics, to reveal the possible and measurable factors related to children with HD. For example, asking children to strengthen or loosen their force applied from a specific digit is a relatively direct way to change the writing pattern. However, in the real world, we could not know if this is necessary or correct to ask them to change the force if we do not know how much force is poor. Therefore, this study provided the real force data such as the AF value, suggested the more critical situation for observing such as pen in air, and proposed the particular digit for observing such as index finger to tentatively bring out the possible and manipulatable factors related to HD. In addition, this study introduced CVF, NFFPS, and FR to explain the force control pattern in children with HD. The study provided a practical method for the teachers and clinicians to assess children’s handwriting problems based on writing kinetics.”.

Please refer to Page 17 lines 291-298; Page 20 lines 353-364.

---

## [Decision Letter · Decision Letter 2]

6 Apr 2022

PONE-D-21-12263R2

Pen-grip Kinetics in Children with and without Handwriting Difficulties

PLOS ONE

Dear Dr. Kuo,

Thank you for submitting your manuscript to PLOS ONE. After careful consideration, we feel that it has merit but does not fully meet PLOS ONE’s publication criteria as it currently stands.  Therefore, we invite you to submit a revised version of the manuscript that addresses the points raised during the review process.

We look forward to receiving your revised manuscript.

Kind regards,

Victor Frak, MD, Ph.D

Academic Editor

PLOS ONE

Journal Requirements:

Reviewers' comments:

Reviewer's Responses to Questions

**Comments to the Author**

1. If the authors have adequately addressed your comments raised in a previous round of review and you feel that this manuscript is now acceptable for publication, you may indicate that here to bypass the “Comments to the Author” section, enter your conflict of interest statement in the “Confidential to Editor” section, and submit your "Accept" recommendation.

Reviewer #1: All comments have been addressed

Reviewer #2: (No Response)

2. Is the manuscript technically sound, and do the data support the conclusions?

Reviewer #1: Yes

Reviewer #2: Partly

3. Has the statistical analysis been performed appropriately and rigorously? 

Reviewer #1: Yes

Reviewer #2: Yes

4. Have the authors made all data underlying the findings in their manuscript fully available?

Reviewer #1: Yes

Reviewer #2: Yes

5. Is the manuscript presented in an intelligible fashion and written in standard English?

Reviewer #1: Yes

Reviewer #2: No

6. Review Comments to the Author

Reviewer #1: I believe that the authors did an excellent job of reviewing and readjusting the manuscript. I consider the article complies Plos One's publishing standards.

Reviewer #2: The authors have addressed earlier comments concerning the results and discussion sections and the overall quality of the manuscript has increased. However, the writing of these two particular sections needs to be improved through better organization of the information.

Results section

• The authors state: “An analysis of covariance (ANCOVA) controlling for age was used to analyze the experimental data to determine the between-group differences in the parameters (the AF on paper of the pen-tip, the TTs, the CVF on paper of the pen-tip, and the FR) and raw BOT-2 scores. A mixed-model ANCOVA including one between factor (group: with HD vs. without HD) and one within factor (role of the digits: thumb vs. index finger vs. middle finger) was performed on the AF on paper, the AF in air, the AF whole task, the CVF whole task, and the Handwriting difficulties and handwriting kinetics NFFPS whole task, with age as a covariate” (pp 12-13, Lines 215-221).

The reader would therefore expect the findings to be described in the same order: the ANCOVA results and then the mixed-model ANCOVA results. However, the description is not clear because the results of the two analyses are described together in the same paragraph. Regarding the ANCOVA, the results concerning “fine motor skills” and “AF on paper” are described correctly; however, the results concerning “TTs”, “CVF on paper of the pen-tip”, and the “FR” are not mentioned (see p.13, lines 227-230). Instead, these results are mentioned under the description of the mixed-model ANCOVA findings (p. 13, lines 229-235).

Furthermore, information concerning the mixed-model ANCOVA results is repeated in the “Group differences” and “Role differences” sub-sections.

Discussion section

• The same organizational problems occur as in the results section. For example, AF-in-air results are addressed on lines 286-300, but suddenly appear again on lines 316-317. Similarly, the interaction Group x Digit on CVF whole task is described on lines 318-321 and appears again on lines 371-373.

• Lines 287-290 are difficult to follow: “The AF-whole task and AF-on paper results revealed statistically significant but less practical significance (p=0.02 and 0.03; effect size ηp2 =0.09 and 0.07) between-group differences for any of the three digits. However, the AF-in air in all digits was practically and significantly (p=0.00; effect size ηp2 =0.18) lower in the HD group”.

• It is suggested that some introductory lines be included in the first paragraph to remind the reader of the objective of the study.

• It is also suggested that the statistical data be removed from the discussion and introduced in the results section instead.

There remain a number of unclear sentences and instances where information could be presented more clearly. The manuscript would therefore benefit from a careful review by a native speaker.

7. PLOS authors have the option to publish the peer review history of their article (what does this mean?). If published, this will include your full peer review and any attached files.

Reviewer #1: **Yes: **Ronaldo Luis da Silva

Reviewer #2: No

---

## [Author Response · Author response to Decision Letter 2]

3 May 2022

Reviewer #1:

I believe that the authors did an excellent job of reviewing and readjusting the manuscript. I consider the article complies Plos One's publishing standards.

Response: We sincerely appreciate the Reviewer’s thorough review and comment.

Reviewer #2: 

The authors have addressed earlier comments concerning the results and discussion sections and the overall quality of the manuscript has increased. However, the writing of these two particular sections needs to be improved through better organization of the information.

Response: We sincerely appreciate the Reviewer’s thorough review and valuable suggestions. The authors revised the manuscript according to your constructive comments. In addition, a native English speaker, who is also a professional proofreader, was invited to help with English editing of the revised manuscript. The point-by-point responses are shown as follows:

Results section

1. The authors state: “An analysis of covariance (ANCOVA) controlling for age was used to analyze the experimental data to determine the between-group differences in the parameters (the AF on paper of the pen-tip, the TTs, the CVF on paper of the pen-tip, and the FR) and raw BOT-2 scores. A mixed-model ANCOVA including one between factor (group: with HD vs. without HD) and one within factor (role of the digits: thumb vs. index finger vs. middle finger) was performed on the AF on paper, the AF in air, the AF whole task, the CVF whole task, and the Handwriting difficulties and handwriting kinetics NFFPS whole task, with age as a covariate” (pp 12-13, Lines 215-221).

The reader would therefore expect the findings to be described in the same order: the ANCOVA results and then the mixed-model ANCOVA results. However, the description is not clear because the results of the two analyses are described together in the same paragraph. 

Response: Thanks for the Reviewer’s constructive comments and correction. The authors have changed the order according to reviewer’s suggestion. The subtitles as “group differences” and “Role differences” were also removed. Please refer to Page 13 line 221 to Page 14 line 248.

The results of the two analyses were described in different paragraph in the revised manuscript. The description was revised as “Table 2 shows the means and SDs for all parameters and fine motor scores.

Table 2 also shows the ANCOVA results on the group differences for the parameters (AFon paper of the pen tip, TTs, CVFon paper of the pen tip and FR) and fine motor scores. For the fine motor skills, scores for ‘fine motor precision’ and ‘fine motor integration’ were significantly lower in the HD group. The AFon paper of the pen tip was significantly lower in the HD group. For the TTs, between-group significance was only found in the TTwhole task. For the CVFon paper of the pen tip, no significance was found between groups. Finally, the FRon paper was significantly lower in the HD group.”. Please refer to Page 13 lines 221-228.

2. Regarding the ANCOVA, the results concerning “fine motor skills” and “AF on paper” are described correctly; however, the results concerning “TTs”, “CVF on paper of the pen-tip”, and the “FR” are not mentioned (see p.13, lines 227-230). Instead, these results are mentioned under the description of the mixed-model ANCOVA findings (p. 13, lines 229-235).

Response: Thanks for the Reviewer’s comments. The authors have changed the order by mentioning the results of ANCOVA first and then the results of mixed-model ANCOVA according to reviewer’s suggestion. Therefore, the results concerning “TTs”, “CVF on paper of the pen-tip”, and the “FR” were now under the description of the ANVOVA. 

The description was revised as “Table 2 shows the means and SDs for all parameters and fine motor scores.

Table 2 also shows the ANCOVA results on the group differences for the parameters (AFon paper of the pen tip, TTs, CVFon paper of the pen tip and FR) and fine motor scores. For the fine motor skills, scores for ‘fine motor precision’ and ‘fine motor integration’ were significantly lower in the HD group. The AFon paper of the pen tip was significantly lower in the HD group. For the TTs, between-group significance was only found in the TTwhole task. For the CVFon paper of the pen tip, no significance was found between groups. Finally, the FRon paper was significantly lower in the HD group.”. Please refer to Page 13 lines 221-228.

3. Furthermore, information concerning the mixed-model ANCOVA results is repeated in the “Group differences” and “Role differences” sub-sections.

Response: Thanks for the Reviewer’s constructive comments and correction. The authors have changed the order by mentioning the results of ANCOVA first and then the results of mixed-model ANCOVA according to reviewer’s suggestion. The subtitles as “group differences” and “Role differences” were also removed.

Therefore, the information concerning the mixed-model ANCOVA results was not repeated in the “Group differences” and “Role differences” sub-sections anymore. Please refer to Page 13 line 221 to Page 14 line 248.

Discussion section

4. The same organizational problems occur as in the results section. For example, AF-in-air results are addressed on lines 286-300, but suddenly appear again on lines 316-317. Similarly, the interaction Group x Digit on CVF whole task is described on lines 318-321 and appears again on lines 371-373.

Response: Thanks for the Reviewer’s constructive comments and correction.

For the AF-in-air results, the description on lines 316-317 has been removed. The description of AF-in-air results have been revised as “Although the differences were found in all pen positions, only the difference of AFin air in all digits between groups was practically significant because its effect size is large enough to be meaningful in practice. The AFin air in all digits was practically and significantly lower in the HD group.”, and “A low CVF represents a static way of grasping a pen in terms of the amount of force. Although not all CVFs showed significant between-group differences, the mean CVF values for the three digits were smaller in the HD group. Our results indicated that the children with HD used a less stable method by which to control the amount of force applied to manipulate the pen during writing.”. Please refer to Page 17 lines 282-285; Please refer to Page 18 lines 311-315.

For the interaction Group x Digit on CVF whole task, the description on lines 318-321 has been removed. The description was revised as “A low CVF represents a static way of grasping a pen in terms of the amount of force. Although not all CVFs showed significant between-group differences, the mean CVF values for the three digits were smaller in the HD group. Our results indicated that the children with HD used a less stable method by which to control the amount of force applied to manipulate the pen during writing.”. Please refer to Page 18 lines 311-315.

5. Lines 287-290 are difficult to follow: “The AF-whole task and AF-on paper results revealed statistically significant but less practical significance (p=0.02 and 0.03; effect size ηp2 =0.09 and 0.07) between-group differences for any of the three digits. However, the AF-in air in all digits was practically and significantly (p=0.00; effect size ηp2 =0.18) lower in the HD group”.

Response: Thanks for the Reviewer’s constructive comments. The authors have revised the description and the statistical data were also removed according to the reviewer’s suggestion. The description was revised as “Although the differences were found in all pen positions, only the difference of AFin air in all digits between groups was practically significant because its effect size is large enough to be meaningful in practice. The AFin air in all digits was practically and significantly lower in the HD group.”. Please refer to Page 17 lines 282-285.

6. It is suggested that some introductory lines be included in the first paragraph to remind the reader of the objective of the study.

Response: Thanks for the Reviewer’s constructive comments. The introductory lines were added in the first paragraph of the Discussion part according to reviewer’s suggestion. The description was revised as “HD is a wildly discussed issue that has received considerable attention from clinical and educational professionals. Previous studies have revealed valuable kinematics related to handwriting performance. However, the reports on children’s handwriting performance by investigating the force directly applied from the digit are still limited. This study discovered the pen-grip kinetics during writing via the FAP system of children with HD compared with those without HD. In addition, the relationship among three digits during writing was discussed to better understand the digits’ role.”. Please refer to Page 16 lines 257-263.

7. It is also suggested that the statistical data be removed from the discussion and introduced in the results section instead.

Response: Thanks for the Reviewer’s constructive comments. The authors have removed the statistical data according to the reviewer’s suggestion and also revised the related description. The description was revised as “Although the differences were found in all pen positions, only the difference of AFin air in all digits between groups was practically significant because its effect size is large enough to be meaningful in practice. The AFin air in all digits was practically and significantly lower in the HD group.”, and “A low CVF represents a static way of grasping a pen in terms of the amount of force. Although not all CVFs showed significant between-group differences, the mean CVF values for the three digits were smaller in the HD group. Our results indicated that the children with HD used a less stable method by which to control the amount of force applied to manipulate the pen during writing.”. Please refer to Page 17 lines 282-285; Please refer to Page 18 lines 311-315.

8. There remain a number of unclear sentences and instances where information could be presented more clearly. The manuscript would therefore benefit from a careful review by a native speaker.

Response: We appreciate the reviewer’s suggestion. A native English speaker, who is also a professional proofreader, was invited to help with English editing of the revised manuscript.

---

## [Decision Letter · Decision Letter 3]

13 Jun 2022

Pen-grip Kinetics in Children with and without Handwriting Difficulties

PONE-D-21-12263R3

Dear Dr. Kuo,

We’re pleased to inform you that your manuscript has been judged scientifically suitable for publication and will be formally accepted for publication once it meets all outstanding technical requirements.

Kind regards,

Victor Frak, MD, Ph.D

Academic Editor

PLOS ONE

Additional Editor Comments (optional):

Dear Doctor Li-Chieh Kuo,

There is still a small observation, which you can complete during the editing process,

Best regards,

V Frak

Reviewers' comments:

Reviewer's Responses to Questions

**Comments to the Author**

1. If the authors have adequately addressed your comments raised in a previous round of review and you feel that this manuscript is now acceptable for publication, you may indicate that here to bypass the “Comments to the Author” section, enter your conflict of interest statement in the “Confidential to Editor” section, and submit your "Accept" recommendation.

Reviewer #2: All comments have been addressed

2. Is the manuscript technically sound, and do the data support the conclusions?

Reviewer #2: Yes

3. Has the statistical analysis been performed appropriately and rigorously? 

Reviewer #2: Yes

4. Have the authors made all data underlying the findings in their manuscript fully available?

Reviewer #2: Yes

5. Is the manuscript presented in an intelligible fashion and written in standard English?

Reviewer #2: Yes

6. Review Comments to the Author

Reviewer #2: The authors of the manuscript entitled “Pen-grip kinetics in children with and without handwriting difficulties” with reference PONE-D-21-12263R3 have responded very well to the various changes suggested. I am happy to recommend the publication of their manuscript in Plos-one.

A minor point:

• p.17, lines 281-285. The sentences “Although the differences were found in all pen positions, only the difference of AFin-air in all digits between groups was practically significant because its effect size is large enough to be meaningful in practice. The AFin-air in all digits was practically and significantly lower in the HD group” are still confusing. I suggest indicating that differences between groups were found to be significant in all pen positions; at least, this is what can be deduced from the p-values in Table 3.

7. PLOS authors have the option to publish the peer review history of their article (what does this mean?). If published, this will include your full peer review and any attached files.

Reviewer #2: No

---

## [Editor Report · Acceptance letter]

16 Jun 2022

PONE-D-21-12263R3 

Pen-grip Kinetics in Children With and Without Handwriting Difficulties 

Dear Dr. Kuo:

I'm pleased to inform you that your manuscript has been deemed suitable for publication in PLOS ONE. Congratulations! Your manuscript is now with our production department. 

Kind regards, 

on behalf of

Dr. Victor Frak 

Academic Editor

PLOS ONE